# A monoclonal antibody that inhibits the shedding of CD16a and CD16b and promotes antibody-dependent cellular cytotoxicity against tumors

Bruna Taciane da Silva Bortoleti [1], Sophia Quasem[1], Stefanie Maurer[1], Xiaoxuan Zhong[1,9], Ruan Pimenta [1], Luiza Ribeiro de Lima Brandão [1], Matthew Hernandez[1], Melanie Fraidenburg[1,10], Pedro Henrique Alves da Silva[1], Raymond Alvarez[2,11], Benjamin K. Chen [1,2], Márcio Augusto Diniz [3], Brian Housman[4], Raja M. Flores[4], Rachel Brody[5], Thomas U. Marron [1,6,7] & Lucas Ferrari de Andrade [1,6,7,8] ✉

CD16a triggers antibody-dependent cellular cytotoxicity (ADCC) and phagocytosis by natural killer (NK) cells and macrophages in anti-tumor immunity. However, CD16a undergoes cleavage by ADAM17 that dampens its anti-tumor immunity. We here develop a monoclonal antibody (F9H4) that binds to CD16a and inhibits its cleavage. F9H4 retains CD16a on the surface of NK cells and macrophages, without triggering or blocking CD16a. F9H4 also binds to and inhibits shedding of CD16b by neutrophils, and inhibits CD16a/b shedding by leukocytes in tumor samples from lung cancer patients. F9H4 promotes ADCC against lung cancer cells that are opsonized by cetuximab, an epidermal growth factor receptor antibody that engages CD16a. F9H4 synergizes with cetuximab to inhibit human lung adenocarcinoma development in immunodeficient mice reconstituted with human NK cells. F9H4 combining with cetuximab also inhibits murine lung carcinoma growth in Fc gamma receptor-humanized mice, and such effect is mediated by NK cells and macrophages. The efficacy of F9H4+cetuximab in lung cancer models is the proof-of-concept for this new approach that promotes anti-tumor functions of Fc-enabled antibodies.

Antibodies provide immunity against infections and cancers, in part through Fcγ receptor recognition of the Fc by leukocytes[1]. CD16a is the Fcγ receptor that triggers antibody-dependent cellular cytotoxicity (ADCC) by natural killer (NK) cells, and works coordinately with other two activating (CD32a and CD64) and one inhibitory (CD32b) Fcγ receptors to trigger antibody-dependent cellular phagocytosis (ADCP) by macrophages[1]. Several cancer therapies consist of antibodies that engage CD16a to trigger cellular immunity, such as cetuximab that

binds and blocks epidermal growth factor receptor (EGFR) in tumor cells and induces ADCC[2]. However, CD16a undergoes ectodomain shedding, which is a post-translational modification performed by A disintegrin and metalloprotease 17 (ADAM17) that releases the CD16a extracellular domain from the cellular surface[3,4]. NK cells shed CD16a in response to stimulations with, for example, protein kinase C agonist and the interleukins 12 and 18[3,5]. The human plasma has soluble CD16a molecules shed by NK cells[6,7]. On the other hand, deletion of the

*ADAM17* gene in NK cells or protease inhibitors stop CD16a shedding and promote ADCC[3,8,9]. However, ADAM17 has multi-substrate specificity, and therefore, inhibitors targeting it could cause pleiotropic effects if administered in vivo[10]. Supporting this notion, ADAM17-deficient mice die at an early stage, and human ADAM17 deficiency, which is rare, causes multiple abnormalities with most of patients dying at infancy or childhood[11–14]. In a phase I/II study with metastatic breast cancer (NCT01254136), a metalloprotease inhibitor with higher specificity for ADAM17 decreased the plasma levels of CD16a and human epidermal growth factor receptor 2 (HER2), which is another target of ADAM17, but the study was terminated because of heterogeneous clinical outcomes presumably caused by unspecific inhibition of cleavage[15]. Therefore, "CD16a shedding" is a target to promote ADCC by NK cells in cancer immunotherapy research but it is limited by the ADAM17's multi-substrate specificity.

An alternative approach would be to genetically engineer NK cells to express a non-cleavable mutant CD16a, characterized by a single amino acid substitution (S197P) near the cleavage site that prevents the shedding and, consequently, promotes ADCC[5,9,16]. Induced pluripotent stem cell (iPSC)-derived NK cells can be engineered to over-express this cleavage resistant CD16a, have remarkable efficacy in pre-clinical cancer models, and transitioned to phase-I clinical trial in patients with cancers (e.g., NCT05395052)[16]. However, these clinical trials have not yet advanced, and conceptually, this cellular therapy approach does not replace endogenous NK cells and macrophages that express and shed CD16a wild type (WT). Therefore, CD16a[S197P]-expressing NK cells help generate the proof-of-concept that CD16a shedding is an immunotherapy target but have translational limitations.

Not only NK cells but also alveolar macrophages shed CD16a in vitro[17]. Furthermore, neutrophils express the CD16b Fcγ receptor that has extracellular domain with higher than 95% amino acid identity with CD16a and regulates the neutrophil responses to immunocomplexes[1]. Neutrophils rapidly shed CD16b upon stimulation with protein kinase C agonist in vitro[18]. Therefore, ectodomain shedding regulates surface CD16a and CD16b (CD16a/b) expressions in NK and myeloid cells and neutrophils.

We here postulate that CD16a/be shedding should be inhibited in a pharmacological manner that targets the substrate and spares the protease. We leverage the high specificity of antibodies to develop the first-in-class monoclonal antibody (mAb) that inhibits CD16a/b shedding through the binding to CD16a/b. This mAb (F9H4) retains CD16a on the NK cell and macrophage surface and CD16b on the neutrophil surface, is not an agonist or blocker, and synergizes with cetuximab to promote ADCC against lung cancer cells and inhibit tumor growth. Therefore, F9H4 inhibits CD16a/b shedding in a highly specific manner and is a new opportunity to enhance the efficacy of tumor cell-opsonizing antibodies for cancer immunotherapy by promoting Fc receptor engagement.

## Results

### Development and validation of F9H4, a mAb inhibitory of CD16a/b shedding

By hybridoma technology whereby Balbc mice were immunized with recombinant human chimeric CD16a full-length extracellular region protein that was fused to human IgG1 Fc, we generated F9H4, a mouse mAb against the human CD16a protein (Fig. 1A). We discovered that F9H4 had the unique property to disrupt the CD16a shedding into supernatants of phorbol myristate acetate (PMA)-stimulated human NK cells that were isolated from the blood of volunteer donors (Fig. 1B; Supplementary Figs. 1 and 2A). For clarification, PMA is a protein kinase C agonist that activates the ADAM17-mediated cleavage of surface proteins[10]. As a consequence of F9H4-mediated inhibition of shedding, NK cells kept CD16a on the cellular surface (Fig. 1C, D and Supplementary Fig. 2B, C). PMA induced expression of the CD69 activation

marker together with CD16a downregulation in NK cells, but the latter was inhibited by F9H4 (Fig. 1E and Supplementary Fig. 2D). F9H4 also inhibited CD16a shedding by human monocyte-derived macrophages (Fig. 1F and Supplementary Fig. 3). Furthermore, F9H4 bound to CD16b and inhibited CD16b shedding by neutrophils (Fig. 1G and Supplementary Fig. 4A, B). F9H4 also inhibited CD16a/b shedding by fresh NK cells, monocytes, and neutrophils that were treated right after isolation of peripheral blood mononuclear cell samples from volunteer donors (Supplementary Fig. 5A–C). To evaluate the alternative hypothesis that F9H4 might inhibit internalization rather than cleavage, we engineered a leukemia cell line to express CD16a[WT] or the non-cleavable mutant (CD16a[S197P]) (Supplementary Fig. 6A). Given the inability of the mutant isoform to undergo cleavage, if F9H4 inhibit such internalization, we would expect that CD16a[S197P] cells would have higher expression levels of CD16a on the cellular surface; however, that was not observed. Rather, F9H4 slightly lowered the CD16a staining in CD16a[WT] cells not treated with PMA and CD16a[S197P] cells with or without PMA likely through modest antigen internalization in vitro, but the cleavage caused a greater downregulation that was inhibited by F9H4 (Fig. 1H and Supplementary Fig. 6B). Since F9H4 was a mouse IgG1 (mIgG1) molecule (Supplementary Fig. 7), we developed a humanized version of F9H4 by sequencing the hybridoma and cloning the variable regions into a vector that enabled expression of recombinant antibodies in mammalian cells. With this approach, we expressed F9H4 with the human IgG1 (hIgG1) Fc and D265A N297A (DANA) mutations. DANA mutations eliminate the binding between Fc and Fcγ receptors or the complement system but they do not abrogate binding to the neonatal Fc receptor[19]. We validated that F9H4 as DANA-mutant hIgG1 bound CD16a (Supplementary Fig. 8A); this binding should be mediated by the variable regions. On the other hand, F9H4 with the DANA mutations did not bind another Fcγ receptor (CD64) (Supplementary Fig. 8B), thus indicating that its Fc was inert. Furthermore, we validated that F9H4-hIgG1-DANA bound CD16a but not CD32a, CD32b/c, or CD64 (supplementary Fig. 8C). F9H4-hIgG1-DANA inhibited CD16a shedding by NK cells (Fig. 1I). Therefore, F9H4 is an anti-CD16a/b mAb that inhibits the shedding of CD16a/b.

### Insights into the molecular mechanisms of action of F9H4

The CD16a extracellular region consists of two immunoglobulin (Ig)-like domains, called D1 and D2. They are connected by a hinge and form hydrogen bonds and Van der Waals interactions by hydrophobic amino acids that cause D1 to bend over D2 at 52° angle[20]. D2 binds the Fc, whereas D1 does not physically interact with the Fc (Fig. 2A)[21]. Using a flow cytometry-based bead assay that analyzed the binding between hIgG1 and CD16a (Supplementary Fig. 9A), we discovered that F9H4 did not obstruct interaction between CD16a and the Fc (Fig. 2B and Supplementary Fig. 9B, C). The clone 3G8 inhibited the binding in a dose-dependent manner (Fig. 2B and Supplementary Fig. 9B, C). For clarification, 3G8 is a known anti-CD16a/b mAb commercially available and with an epitope in the Fc recognition region[22]. F9H4 did not compete with 3G8 (Supplementary Fig. 10). Furthermore, 3G8 is an agonistic antibody[23]. By using an engineered cell line that served as CD16a engagement reporter, we confirmed that 3G8 was an agonist. In contrast, F9H4 was non-agonist (Fig. 2C). Since F9H4 did not block CD16a or compete with 3G8 and was not an agonist, we hypothesized that F9H4 binds D1, which is the Ig-like domain that does not bind the Fc (Fig. 2A). To test this hypothesis, we generated three recombinant proteins that consisted of full-length CD16a (D1 + D2), D1, or D2. We detected the binding of F9H4 to full-length CD16a and, to a lesser extent, D1; in contrast, there was little to no binding of F9H4 to D2 (Fig. 2D, E). Given that the binding pattern for D1 changed to an *S*-curve that does not plateau like that for full-length CD16a, it is likely that most but not all the F9H4 epitopes were in D1.

The CD16a cleavage site is in the stalk, which is a linear peptide in-between D2 and the transmembrane domain[5,24]. Since F9H4 bound D1,

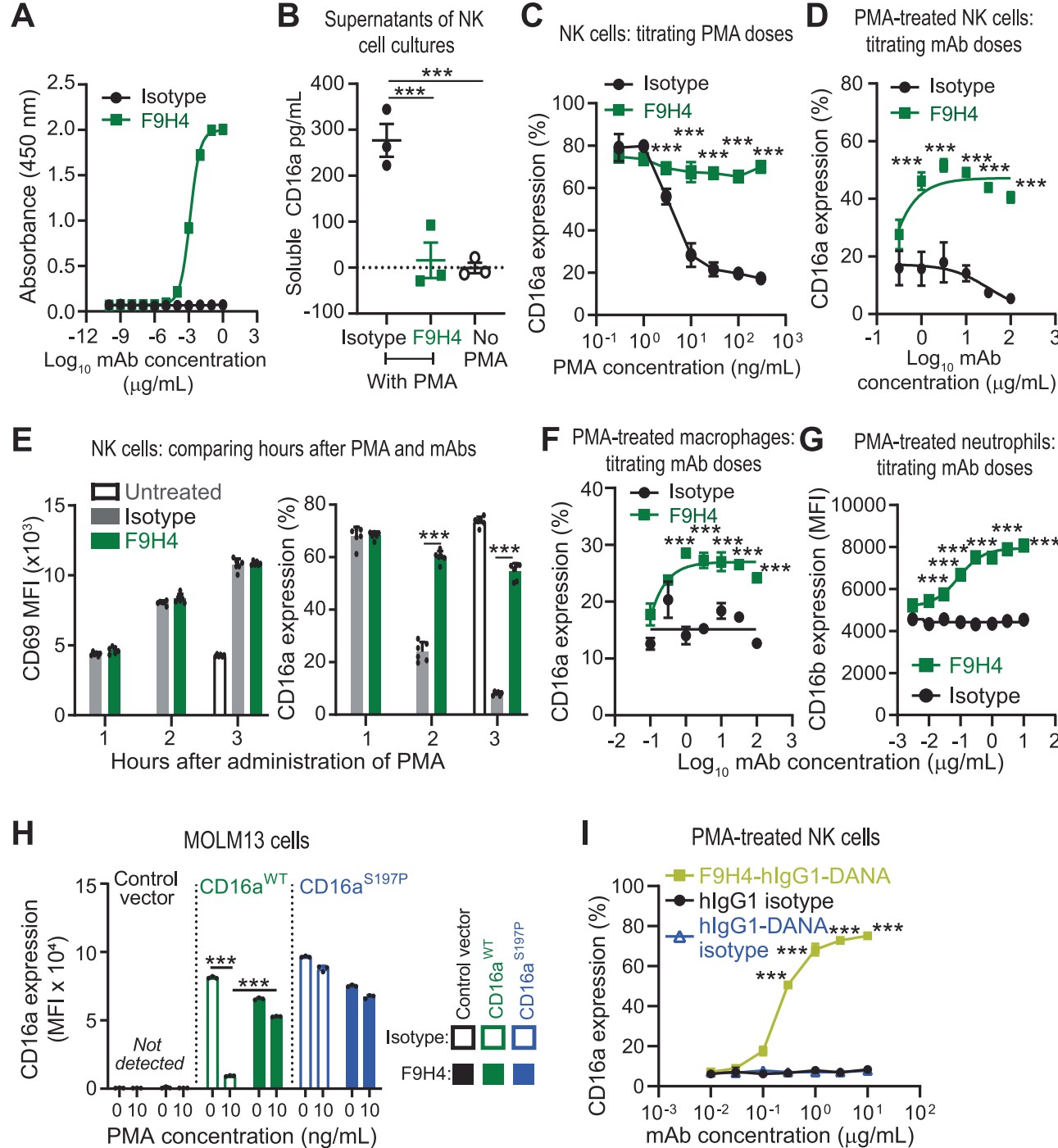

**Fig. 1 | F9H4 is an anti-human CD16a/b antibody that inhibits CD16a/b shedding. A** ELISA for analysis of the binding of F9H4 to recombinant human CD16a protein. **B** Detection of soluble CD16a shed by NK cells. The surface proteins of NK cells were covalently linked to biotin. Subsequently, NK cells were treated for 4 h with PMA to induce the shedding of biotinylated CD16a, which in turn was captured in ELISA plates by immobilized 3G8. Biotinylated CD16a was detected with peroxidase-labeled streptavidin. The negative values were caused by interpolation of absorbance values below detection limit of the standard curve. **C–I** Surface CD16a/b was analyzed by flow cytometry using phycoerythrin (PE)-labeled 3G8. NK cells were treated with 5 ng/ml PMA plus the indicated antibodies for 4 h (**C**) or for 1 h with 5 µg/mL antibodies plus the indicated concentration of PMA (**D**), followed by analyses of surface CD16a by flow cytometry. **E** NK cells were treated with 10 µg/mL of antibodies and 10 ng/mL PMA for the indicated times followed by analyses of CD69 and CD16a expressions by flow cytometry. CD16a/b-shedding assays with human primary monocyte-derived macrophages (**F**) and neutrophils (**G**) that were treated with PMA. **H** The indicated MOLM13 cell lines were treated for 1 h with the indicated antibodies ± PMA, followed by analyses of CD16a expressions by flow cytometry. **I** NK cells that were treated with 30 ng/mL PMA to induce CD16a shedding and the indicated antibodies. Data represent three independent experiments (**A–I**), are mean ± standard deviation (**B, C, E, H, I**) or standard error (**D, F, G**) of triplicates (**B, C, G, H, I**) or four replicates (**D, F**) except for one data point in (**D**) "F9H4 at 3 µg/mL" that is triplicate or six replicates (**E**), and were analyzed by non-linear regression (**A, D, F, G**) and two-sided Rank-Based Multiple Test Procedures and Simultaneous Confidence Intervals (**B–I**). ***p < 0.001. MFI = mean fluorescence intensity (**E, G, H**).

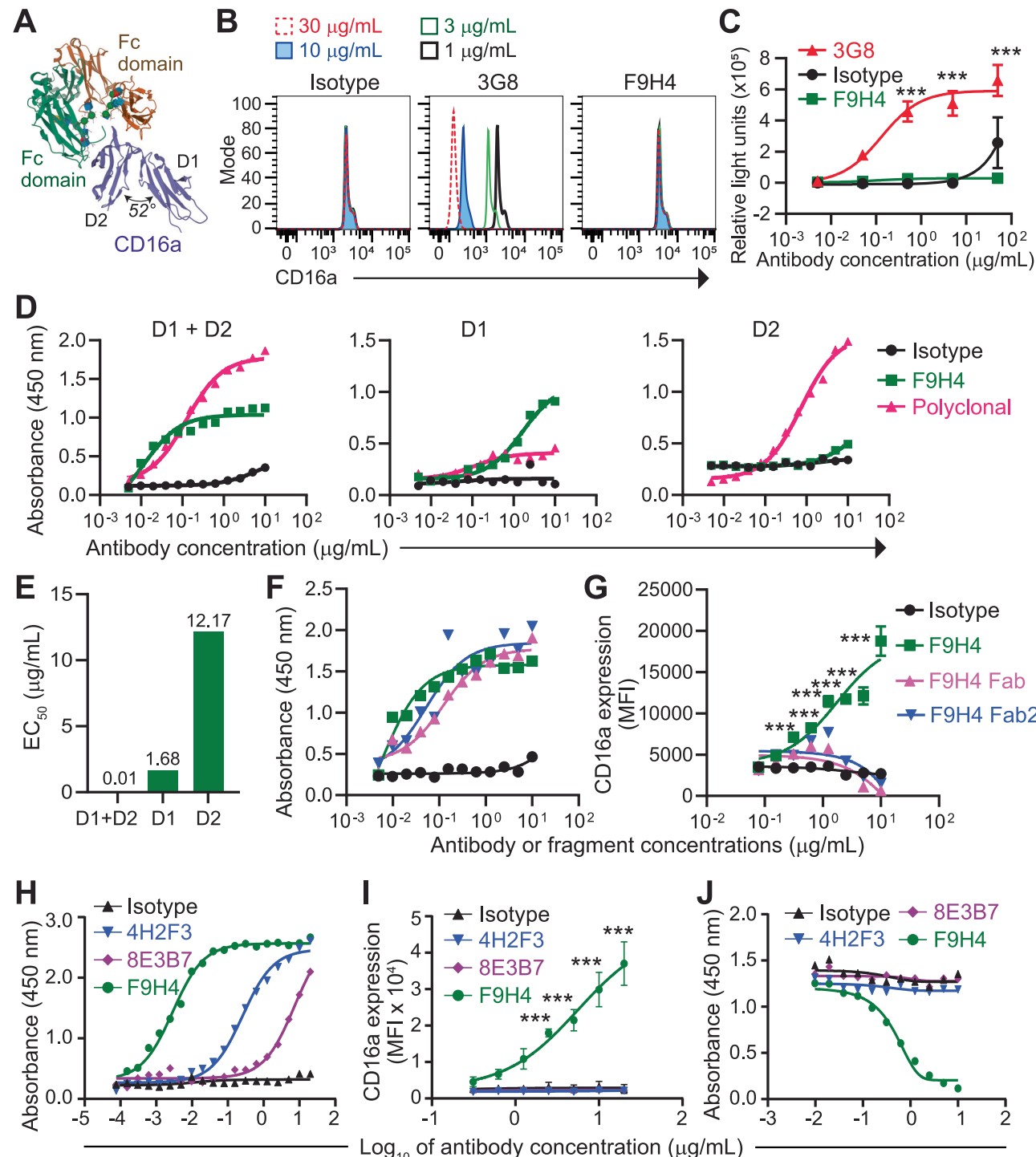

**Fig. 2 | Insights into the molecular mechanism of action of F9H4. A** Crystal structure of CD16a bound to Fc domain of hIgG1. Protein Data Bank 5YC5. **B** hIgG1-coated beads were incubated with 1 μg/mL biotinylated CD16a plus the indicated antibodies, followed by incubation with PE-labeled streptavidin and analysis by flow cytometry. See also Supplementary Fig. 9B. **C** Jukart cells were engineered to express human CD16a and luciferase in response to the triggering of CD16a. The cells were treated overnight with antibodies, followed by analysis of luciferase activity in microplate plate reader. **D** Binding of antibodies to full length CD16a (indicated as "D1 + D2"), D1, or D2, as analyzed by ELISA. The indicated proteins were immobilized in multi-well plates, followed by incubation with the indicated antibodies. **E** The half maximal effective concentration ($EC_{50}$) values of the data in D, for F9H4. **F, G** F9H4 was digested with ficin to generate Fab and Fab2, which were validated for CD16a binding by ELISA with secondary antibody against murine Fab (**F**). CD16a shedding assay with the F9H4 fragments or controls (isotype and F9H4,

which are undigested). MOLM13-CD16a^WT cells were treated for 1 h with 10 μg/mL PMA and the indicated antibodies or fragments, followed by CD16a expression analyses through flow cytometry (**G**). **H, I** ELISA for the reactivity of the indicated mAbs to human CD16a protein (**H**). CD16a shedding assay with MOLM13-CD16a^WT cells (**I**). The cells were treated for 1 h with 10 μg/mL PMA and the indicated mAbs, followed by CD16a expression analyses by flow cytometry. **J** F9H4 does not compete with 4H2F3 and 8E3B7 for CD16a binding. ELISA for the reactivity of the indicated antibodies to full length CD16a protein. The competition assay was done with the humanized version of F9H4, which compete against the murine version that serves as positive control. Data represent three (**B, D–J**) or two (**C**) independent experiments, are mean ± standard deviation (**C, I**) or standard error (**G**) of triplicates (**C, G, I**), and were analyzed by non-linear regression (**C, D, F–J**) and two-way ANOVA with Bonferroni's test (**C, G, I**). ***$p < 0.001$. MFI = mean fluorescence intensity (**G, I**).

which is the membrane distal domain, it did not bind to the cleavage site. Notably, antibodies are relatively large molecules with a molecular weight of approximately 150 kDa. In contrast, the CD16a extracellular region has approximately 50 kDa, thus being three times smaller than antibodies[25]. For these reasons, we hypothesized that F9H4 spatially blocks access to the cleavage site but without binding it. To test this hypothesis, we reduced the F9H4's molecular weight through enzyme-mediated fragmentation. Ficin is a protease that digests the Fc to generate fragment antigen-binding (Fab) and Fab2, which have molecular weights of approximately 50 and 110 kDa, respectively[26]. We used ficin to separate the Fab and Fab2 from the Fc of F9H4. We validated that the F9H4 Fab and Fab2 bound CD16a (Fig. 2F). On the other hand, the F9H4 Fab and Fab2 did not inhibit the CD16a shedding by the engineered leukemia cell line in response to PMA (Fig. 2G). Therefore, the mechanism by which F9H4 inhibits CD16a shedding involved not only its epitope but also its molecular size.

F9H4 was developed through mouse immunization with full-length CD16a extracellular region (D1 + D2). Since F9H4 bound D1, we generated two additional mAbs (4H2F3 and 8E3B7) through mouse immunization with recombinant D1 protein. 4H2F3 and 8E3B7 bound full-length CD16a, but with lower affinity compared to F9H4 (Fig. 2H). 4H2F3 and 8E3B7 did not inhibit the CD16a shedding by PMA-treated leukemia cells that were engineered to express CD16a^WT (Fig. 2I). Despite being D1-targeting mAbs, 4H2F3 and 8E3B7 did not compete with F9H4 for the binding to CD16a (Fig. 2J). These results highlight the uniqueness of F9H4, which was raised against full-length CD16a and bound preferentially to D1; however, the data with 4H2F3 and 8E3B7, generated through mouse immunization with D1, indicated that not any D1-targeting antibody would inhibit the CD16a cleavage.

### F9H4 promoted human NK cell effector functions against cetuximab-opsonized human lung cancer cells

NK cells express only CD16a and thereby no other Fcγ activating receptor might compensate for the CD16a loss by shedding. In contrast, macrophages express CD16a, CD32a, and CD64 that trigger antibody-dependent cellular phagocytosis, whereas the role of CD16b in neutrophils is not well delineated yet[1]. For these reasons, we hypothesized that F9H4 promotes NK cell-mediated ADCC as a consequence of CD16a-shedding inhibition. We used a human lung adenocarcinoma cell line (A549) that was opsonized with cetuximab as a model for NK-mediated ADCC, because cetuximab is a hIgG1 anti-EGFR antibody capable of binding CD16a and enabling ADCC[2]. F9H4 increased the activities of NK cell degranulation, interferon-γ production, and cytotoxicity against cetuximab-opsonized A549 (Fig. 3A–C and Supplementary Fig. 11A–I). Notably, F9H4 did not trigger NK cell degranulation in the absence of CD16a engagement by cetuximab-opsonized A549 (Fig. 3D). F9H4 did not increase NK cell proliferation in vitro (Supplementary Fig. 12A, B). Interestingly, A549 induced CD16a downregulation by NK cells, presumably through cleavage, and that was promoted by cetuximab, whereas F9H4 significantly inhibited the downregulation in both conditions (Fig. 3E, F). NK cells alone had slight decrease (7.5%) in the levels of surface CD16a when treated with F9H4, indicating some degree of antigen internalization. In comparison, the CD16a downregulations presumably by shedding and induced by A549 and A549+cetuximab were of greater degrees (33% and 78%, respectively) and inhibited by F9H4 (Fig. 3E, F). The small surface CD16a downregulation by F9H4 was probably caused by internalization because it did not occur in NK cells that were incubated cold, but the downregulation by PMA, which only occurs at 37 °C, was to a greater extent and inhibited by F9H4 (Supplementary Fig. 13A, B). Therefore, F9H4 promotes ADCC against cetuximab-opsonized A549 primarily by inhibiting the CD16a cleavage that occurs upon the encounter with the target cells.

We also analyzed the phagocytosis of cetuximab-opsonized A549 cells by macrophages. Cetuximab slightly induced ADCP and that was

not further promoted by F9H4 (Supplementary Fig. 14A). Furthermore, we combined F9H4 with rituximab, which is an anti-CD20 antibody. F9H4 increased NK cell degranulation against rituximab-opsonized Raji lymphoma cells but not against trastuzumab-opsonized SKOV3 ovarian cancer cells (Supplementary Fig. 14B, C). Furthermore, F9H4 slightly increased macrophage-mediated ADCP against Raji cells but not against the other two lymphoma cell lines that were treated with rituximab (Supplementary Fig. 14D). Therefore, we could document the ability of F9H4 to promote the Fc effector functions of rituximab against Raji cells.

The transition to in vivo experiments required the use of genetically modified mouse strains because the Fcγ receptor biology differs between mice and humans[1]. hIL-15 NOG mice are immunodeficient and constitutively express human interleukin 15 to enable long-term engraftment of human NK cells (Supplementary Fig. 15A)[27]. Intravenous injection of A549 resulted in metastases in the lungs of hIL-15 NOG mice, and the reconstitution with human NK cells 7 days before A549 inoculation, and to a lesser extent 10 days after it, inhibited the metastasis development (Fig. 3G). In this model, F9H4 only targeted human NK cells because they expressed human CD16a, whereas murine NK cells were absent, and endogenous macrophages and neutrophils expressed murine Fcγ receptors. We discovered that treatment with F9H4 + cetuximab decreased the number of tumors in the lungs after intravenous inoculation of A549, whereas F9H4 or cetuximab alone had no significant impact on this model (Fig. 3H, I, supplementary Fig. 15B). Therefore, F9H4 synergizes with cetuximab to inhibit the formation of tumors in the lungs of immunodeficient mice that are reconstituted with human NK cells.

### F9H4 + cetuximab inhibited tumor growth in immunocompetent mice that expressed human Fcγ receptors

To complement the human NK cell and lung cancer model that was shown above, we used Fcγ receptor-humanized mice (hFcR mice) that are immunocompetent and express only human Fcγ receptors[28]. We discovered that the plasmas of hFcR mice had detectable levels of soluble CD16a/b. Interestingly, mice that were inoculated intravenously with a melanoma cell line that formed metastases in the lungs (B16F10) or an acute myeloid leukemia cell line that engrafted in the blood and bone marrow (C1498-MICB), or inoculated subcutaneously with a mouse lung cancer cell line (LLC1) had higher serum CD16a/b levels (Fig. 4A). These results revealed that the host immunity (presumably NK and myeloid cells) shed CD16a/b in response to cancer. To test the hypothesis that F9H4 inhibits tumor growth, we engineered LLC1 to express human EGFR (hEGFR) to allow the labeling by cetuximab (Supplementary Fig. 16A). For clarification, LLC1 is a mouse lung carcinoma cell line syngeneic in C57BL/6 mice, which is the background of hFcR mice[28,29]. Intravenous injection of LLC1-hEGFR did not result in the formation of tumors in the lungs of hFcR mice (not shown). However, LLC1-hEGFR cells developed into solid tumors after subcutaneous inoculation, and treatment with the humanized version of F9H4 in combination with cetuximab slowed tumor growth. Interestingly, F9H4 alone inhibited tumor growth, but to a lesser extent compared to the combination with cetuximab, presumably because hFcR mice were immunocompetent and might generate endogenous antibodies against LLC1-hEGFR. The therapeutic effects in both antibody treatment groups were characterized by tumor size stabilization —the tumors did not grow further out even after several weeks (Fig. 4B). Furthermore, the mIgG1 version of F9H4 synergized with cetuximab to inhibit LLC1-hEGFR tumor growth in hFcR mice. The mIgG1 version of F9H4 also inhibited tumor growth when administered alone, but such effects were to a lesser extent than F9H4 + cetuximab (Fig. 4C). In these experiments, the mice were euthanized three weeks after tumor cell inoculation to enable analyses of blood and tumor-infiltrating leukocytes. Cetuximab induced CD16a downregulation by blood NK cells, and that was prevented by F9H4 (Fig. 4D).

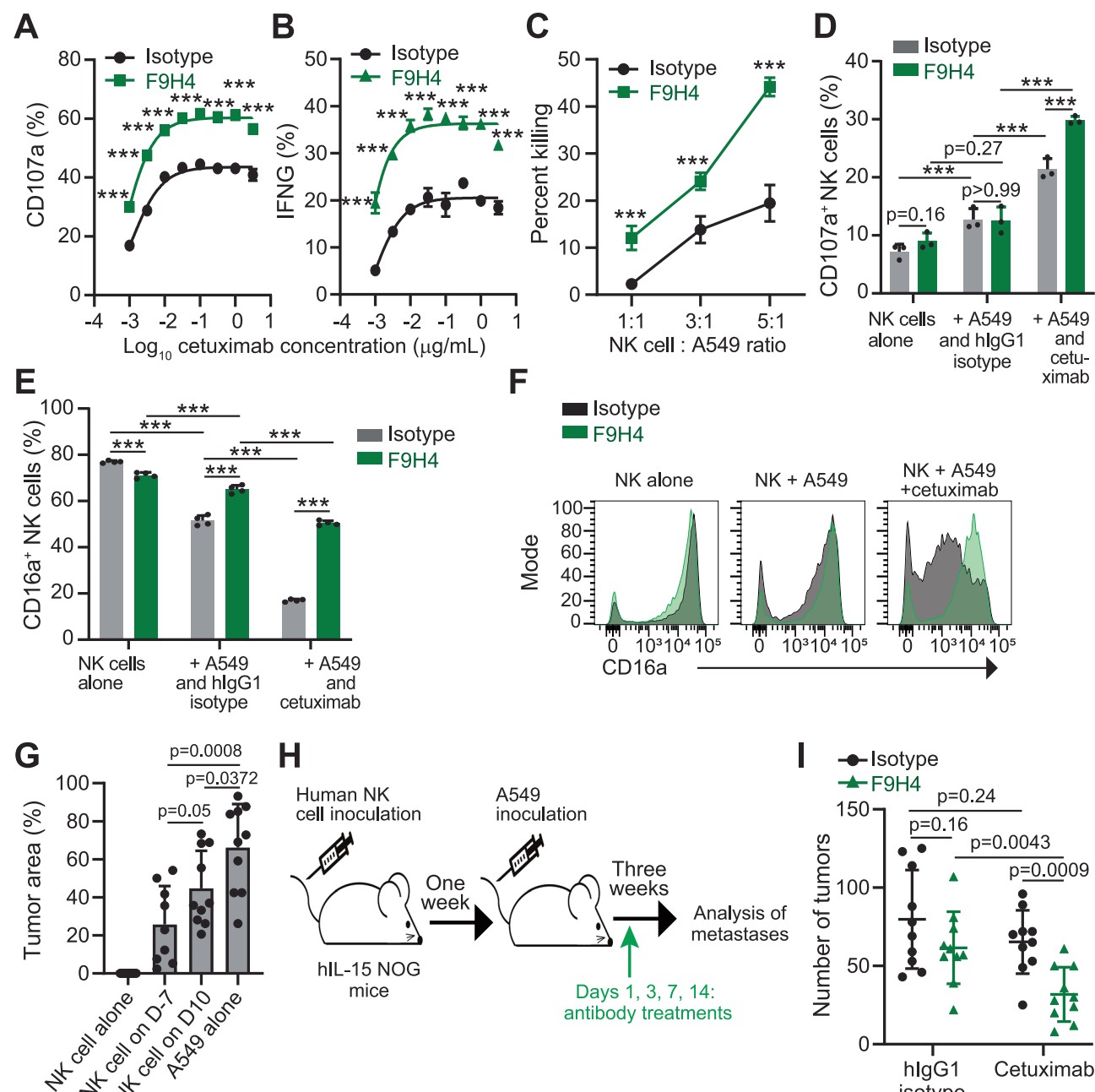

**Fig. 3 | F9H4 + cetuximab promote ADCC by human NK cells and inhibit metastases. A–F** NK cells were co-cultured for 4 h with A549 cells at 1:1 effector-to-target ratio (**A**, **B**, **D–F**) or as indicated (**C**) in the presence of the indicated antibodies, followed by flow cytometry analyses. NK cells were identified as human CD45+ alive single cells. **A** Analysis of CD107a externalization by NK cells. **B** Analysis of intracellular interferon-γ (IFNG) in NK cells. **C** Analysis of A549 cells killed by NK cells, based on the labeling with 7-AAD and correction, by subtraction, for dead A549 cells in the absence of NK cells. NK cells were gated out by using CD56 as marker. **D** CD107a externalization assay with additional control groups. **E**, **F** Expression of surface CD16a in NK cells after co-culture with A549 (or NK cell alone) in the presence of the indicated antibodies. The histograms in (**F**) represent the data in (**E**). **G–I** The A549 lung cancer model in mice reconstituted with human NK cells. **G** Male 6–8 week old mice were inoculated intravenously with $1 \times 10^6$ NK cells from the blood of volunteer donors, on days (D) −7 or 10 relative to A549

inoculation. Analyses of A549 metastases were done by histopathology on day 21. **H** A cartoon illustrating the experimental details. **I** Male 6–8-week old mice were inoculated intravenously with $1 \times 10^6$ human NK cells from the blood of volunteer donors. One week later, the mice were inoculated intravenously with $1.5 \times 10^6$ A549 cells. On days 1 and 2, and once per week after A549 inoculation, the mice were treated with 0.1 mg of each one of the indicated antibodies. Analyses of tumors were done on day 21 by histopathology. Data represent three (**A–F**) or are pooled of two (**G**, **I**) independent experiments, and are mean ± standard deviation (**A**, **B**, **D**, **E**, **G**, **I**) or standard error (**C**) of triplicates (**A–E**) or, in **G**, $n = 9$ in "NK cell on D-7" or $n = 10$ in all other groups in (**G**) and (**I**). Each dot represents one mouse (**G**, **I**). Data were analyzed by two-sided rank-based multiple test procedures and simultaneous confidence intervals (**A–E**), or two-sided unpaired Student's $t$ test (**G**, **I**). *$p < 0.05$, **$p < 0.01$, ***$p < 0.001$.

In contrast, the CD16a expression levels were lower in tumor-infiltrating NK cells from mice that were treated with F9H4 + cetuximab (Fig. 4E). These results lead us to hypothesize that the anti-tumor activity of F9H4 + cetuximab is NK cell mediated in this model. We depleted NK cells in vivo using anti-NK1.1 antibody prior to tumor

cell inoculation and treatment with F9H4 + cetuximab. As shown in Fig. 4F, NK cells were required for optimal tumor growth suppression by F9H4 + cetuximab in the LLC1-hEGFR model in hFcR mice. On the other hand, cetuximab did not induce CD16a downregulation in blood monocytes and tumor-infiltrating macrophages downregulated CD16a

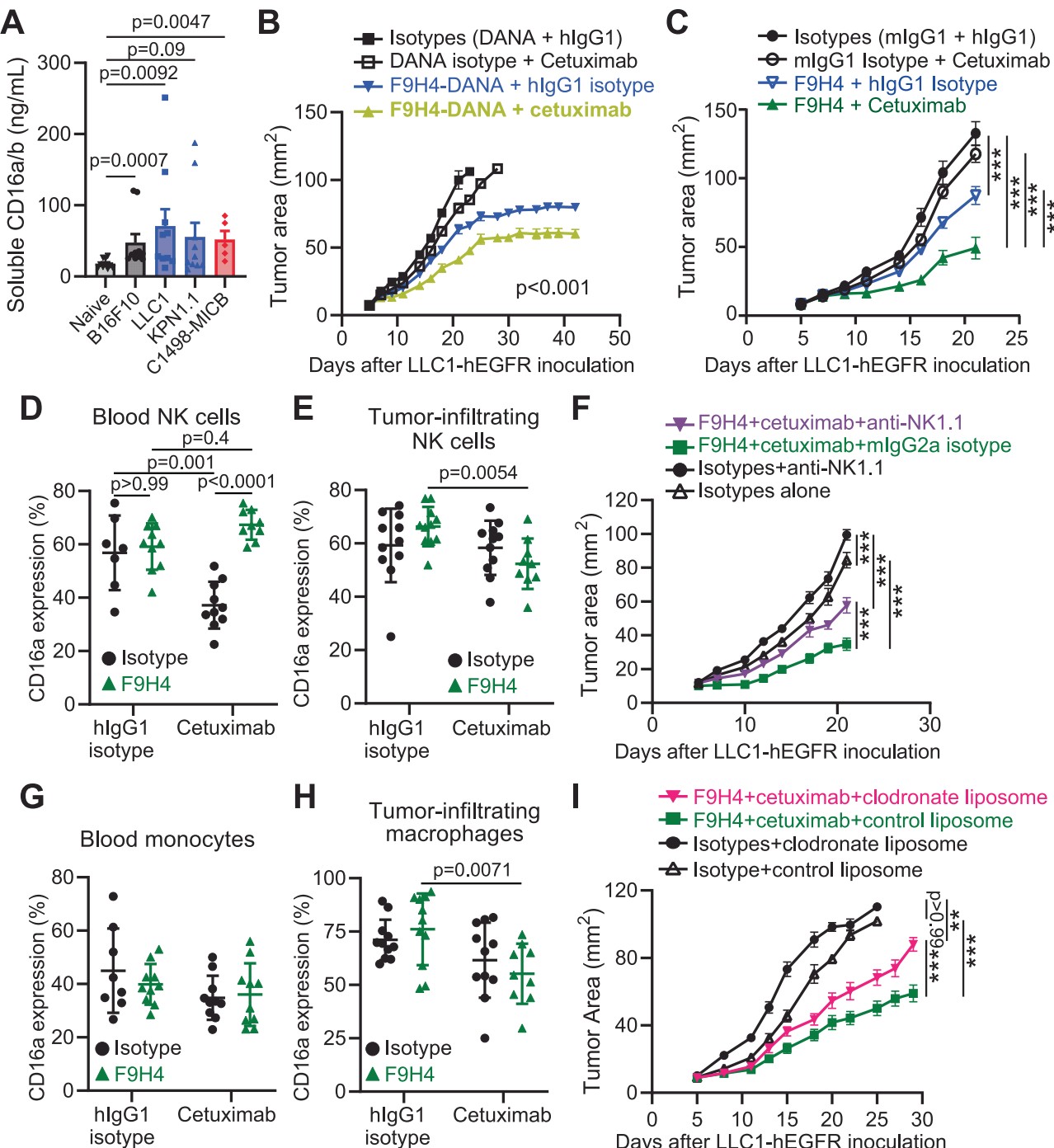

in mice that were treated with F9H4 + cetuximab (Fig. 4G, H). However, macrophage depletion, through intratumoral injections of clodronate liposomes, promoted tumor growth in isotype or F9H4 + cetuximab treated mice (Fig. 4I). Therefore, F9H4 + cetuximab inhibit tumor growth in immunocompetent mice that express human Fcγ receptors, and this activity requires endogenous NK cells and macrophages for optimal efficacy and is associated with higher expression levels of surface CD16a in blood NK cells but downregulation of CD16a in intratumoral NK cells and macrophages.

We also analyzed soluble CD16a/b molecules shed in the blood circulation, by a sandwich ELISA that was described[3,5,18]. hFcR mice that were inoculated with LLC1-hEGFR and treated with F9H4 + cetuximab had higher concentrations of soluble CD16a/b in the sera (Supplementary Fig. 16B). Since F9H4 inhibits the CD16a/b

shedding, we hypothesized that such an increase in serum soluble CD16a/b levels is caused by the F9H4 binding to pre-existing soluble CD16a/b molecules that are generated by cleavage. To test this hypothesis, we used F9H4 to capture soluble CD16a shed in supernatants from the leukemia cell line that was engineered to express CD16a[WT]; F9H4 captured in ELISA plates soluble CD16a shed by this cell line in vitro (Supplementary Fig. 16C). These results led us to ask if soluble CD16a/b molecules that are generated by cleavage bind the Fc. Through a flow cytometry bead assay, we detected soluble CD16a/b in human plasma only when the beads were coated with an antibody to capture CD16a/b, but not when they were coated with hIgG1 (Supplementary Fig. 17A, B). Therefore, F9H4 also binds soluble CD16a/b molecules, which lose the Fc binding activity to WT hIgG1 after cleavage.

**Fig. 4 | F9H4 + cetuximab inhibit tumor growth in immunocompetent mice.**
**A** Female adult hFcR mice were inoculated intravenously with B16F10 or C1498-MICB, or inoculated subcutaneously with LLC1 or KPN1.1, or were not inoculated (Naïve). Analyses of serum CD16a/b were done by ELISA 2 weeks (for the B16F10 model) or 3 weeks (for the C1498-MICB, LLC1, and KPN1.1 models) after cancer cell line inoculations. $N = 10$ mice per group, except in C1498-MICB that is with $n = 5$ mice. **B–I** Male adult hFcR mice were inoculated subcutaneously with $1.5 \times 10^6$ LLC1-hEGFR cells. Tumors were measured by digital caliper (**B, C, F, I**). Treatments with the indicated versions of F9H4, cetuximab, and isotype controls (0.1 mg/antibody/per mouse) were done on days 5, 6, and once per week. Treatments with anti-NK1.1 and mIgG2a isotype were done on days −1, 0, and once per week relative to tumor cell inoculation (**F**). Treatments with control or clodronate liposomes were on days 5, 6, and once per week (**I**). In **B**, DANA isotype + cetuximab $n = 12$, Isotype (DANA + hIgG1) $n = 11$, F9H4-DANA + cetuximab $n = 12$, F9H4-DANA + hIgG1 isotype $n = 12$. In **C**, Isotype + hIgG1 isotype $n = 11$, Isotype + cetuximab $n = 12$, F9H4 + hIgG1 isotype $n = 12$, F9H4 + cetuximab $n = 12$. In **F**, F9H4 + cetuximab + anti-NK1.1 $n = 10$, F9H4 + cetuximab + mIgG2a isotype $n = 10$, Isotypes + anti-NK1.1 $n = 10$, Isotypes alone $n = 9$. In **I**, F9H4 + cetuximab + clodronate liposome $n = 17$, F9H4 +

cetuximab + control liposome $n = 15$, Isotypes + clodronate liposomes $n = 16$, Isotype + control liposome $n = 14$. Percentage of CD16a expression in blood NK cells (**D**) and tumor-infiltrating NK cells (**E**) by flow cytometry. hFcR were subjected to the LLC1-hEGFR model and antibody treatments. On day 21, mice were euthanized and tumor and blood processed for analyses. In **D**, Isotype + hIgG1 isotype $n = 7$, Isotype + cetuximab $n = 9$, F9H4 + hIgG1 isotype $n = 10$, F9H4 + cetuximab $n = 9$. In **E**, Isotype + hIgG1 isotype $n = 10$, Isotype + cetuximab $n = 10$, F9H4 + hIgG1 isotype $n = 11$, F9H4 + cetuximab $n = 9$. Percentage of CD16a expression in blood monocytes (**G**) and tumor-infiltrating macrophages (**H**) by flow cytometry. hFcR mice were under the LLC1-hEGFR model as above. Analyses on day 21. In **G**, Isotype + hIgG1 isotype $n = 8$, Isotype + cetuximab $n = 10$, F9H4 + hIgG1 isotype $n = 10$, F9H4 + cetuximab $n = 10$. In **H**, Isotype + hIgG1 isotype $n = 11$, Isotype + cetuximab $n = 11$, F9H4 + hIgG1 isotype $n = 11$, F9H4 + cetuximab $n = 9$. Data are mean ± standard error (**A–C**, **F**, **I**) or standard deviation (**D, E, G, H**). Each dot represents one mouse (**A, D–F, G, H**). Data represent (**A**) or are pooled of (**B–I**) two independent experiments and were analyzed by two-tailed Mann–Whitney test (**A**) or two-way ANOVA with Bonferroni's test (**B–I**).

## F9H4 also synergized with necitumumab in vivo and an Fc-enhanced version of cetuximab in vitro

Since CD16a binds the hIgG1 Fc, which is conserved, we hypothesized that F9H4 is applicable for combination with other antibodies beyond cetuximab. We analyzed three other antibodies that target EGFR, are hIgG1, and have been tested in patients: (1) Zalutumumab, tested in phase-1 trial with non-small cell lung cancer (NSCLC) patients (NCT00460551); (2) Nimotuzumab, tested in phase-3 trial with head and neck cancer patients (e.g., NCT00957086); and (3) Necitumumab, approved by the Food and Drug Administration for squamous NSCLC[30–32]. To minimize the number of mice required for such a large experiment, we removed the intermediate treatment groups by comparing "F9H4 as hIgG1-DANA plus hIgG1 isotype" with "F9H4 as hIgG1-DANA plus each one of the EGFR antibodies", in addition to the "negative control (hIgG1 and DANA isotypes)" and "positive control (F9H4-hIgG1-DANA plus cetuximab)". We found that F9H4-hIgG1-DANA synergized with necitumumab to inhibit LLC1-hEGFR tumor growth, but zalutumumab and nimotuzumab did not provide a benefit when compared to F9H4-hIgG1-DANA alone (Fig. 5A). By ELISA we found that nimotuzumab was a weaker binder to EGFR, compared to the binding by zalutumumab, cetuximab, or necitumumab (Supplementary Fig. 18A). We also found that zalutumumab was a weaker binder to LLC1-hEGFR cells, compared to the binding by cetuximab (Supplementary Fig. 18B). Therefore, F9H4-hIgG1-DANA synergized with not only cetuximab but also necitumumab to inhibit LLC1-hEGFR tumor growth in hFcR mice.

ADCC is controlled by the binding affinity of CD16a to the Fc. The higher the binding affinity, the more the ADCC[1]. We applied Fc engineering to cetuximab whereby we introduced the G236A, A330L, and I332E (GAALIE) mutations to increase the binding affinity to CD16a. These GAALIE mutations are known to increasing the binding affinity of hIgG1 to all Fcγ activating receptors but not to CD32b, which is the Fcγ inhibitory receptor[1]. Consequently, neutralizing antibodies with GAALIE mutations provided greater immune-mediated protection against influenza virus infection in mouse models[33]. We validated that cetuximab-GAALIE bound EGFR (Supplementary Fig. 18C). Furthermore, the binding affinity was improved for CD16a^F176, whereas both cetuximab versions had similar binding affinity to CD16a^V176 (Supplementary Fig. 18D). For clarification, CD16a^V176 is a polymorphic variant known to have higher binding affinity to hIgG1[34]. F9H4 bound to these two CD16a polymorphic variants (Supplementary Fig. 19). We applied these antibodies to ADCC assays and found that the effects were variable against four different cell lines of human NSCLC. In general, depending on the cell line, the NK cell degranulation and interferon-γ production achieved the maximum possible in the combinations of F9H4 with cetuximab-GAALIE (Fig. 5B, C). Therefore, the inhibition of

CD16a shedding and higher binding affinity, i.e., F9H4 and GAALIE, respectively, complement each other to maximize the stimulation of NK cell effector functions.

EGFR expression is also common in colorectal cancer cells (CRC) and we identified two human CRC cell lines that had contrasting patterns of expression: EGFR^high HCT116 and EGFR^intermediate RKO (Supplementary Fig. 20A). Cetuximab, cetuximab-GAALIE, F9H4 plus cetuximab, and F9H4 plus cetuximab-GAALIE all increased the NK cell degranulation against HCT116. Such effect was superior for F9H4 plus cetuximab compared to cetuximab alone, but F9H4 did not further increase the degranulation with cetuximab-GAALIE against HCT116. In contrast, none of these mAbs promoted NK cell degranulation against the EGFR^intermediate RKO cell line (Supplementary Fig. 20B). Furthermore, we combined F9H4 with the mAb 7C6-hIgG1 in metastasis and subcutaneous tumor models. For clarification, 7C6-hIgG1 binds to and inhibits the shedding of major histocompatibility complex class I polypeptide-related sequence A and B (MICA/B), which are surface proteins expressed by tumor cells in response to stress pathways. MICA/B are recognized by the NK group 2 member D (NKG2D) receptor that triggers NK cell effector functions, but MICA/B shedding causes immune escape. 7C6-hIgG1 stops MICA/B shedding, and consequently it promotes NKG2D recognition and trigger ADCC[35]. Here, we inoculated hFcR mice with B16F10 cells that were engineered to express human MICA, and mAb treatments were administered starting 1 day later. Analyzes on day 14 revealed that 7C6-hIgG1 alone and F9H4 alone greatly inhibited the formation of tumors in the lungs and to a similar extent (Supplementary Fig. 21A). On the other hand, F9H4 plus 7C6-hIgG1 increased the absolute numbers of blood NK cells (Supplementary Fig. 21B). Furthermore, CD16a expression was higher in blood NK cells from mice that received F9H4, 7C6-hIgG1, or both, whereas NKG2D expression was similar among the mAb treatment groups (Supplementary Fig. 21C, D). We also combined these two mAbs in the LLC1 subcutaneous model, in which LLC1 cells were engineered to express human MICA. Similar to the metastasis model, we observed inhibition of tumor growth by the antibody combination and each mAb alone (Supplementary Fig. 21E, F). Collectively, these results did not formally demonstrate synergism between F9H4 and 7C6-hIgG1 in inhibiting tumor growth, but it remains possible that the applicability of F9H4 would not be restricted to EGFR antibodies in lung cancer given that CD16a binds to the Fc, which is a constant region of antibodies.

## Insights into the clinical significance of CD16a/b shedding

We analyzed tumor and paired normal lung tissues from untreated patients with early-stage NSCLC who underwent surgery as the standard-of-care. We applied these specimens to 4-h tissue explants,

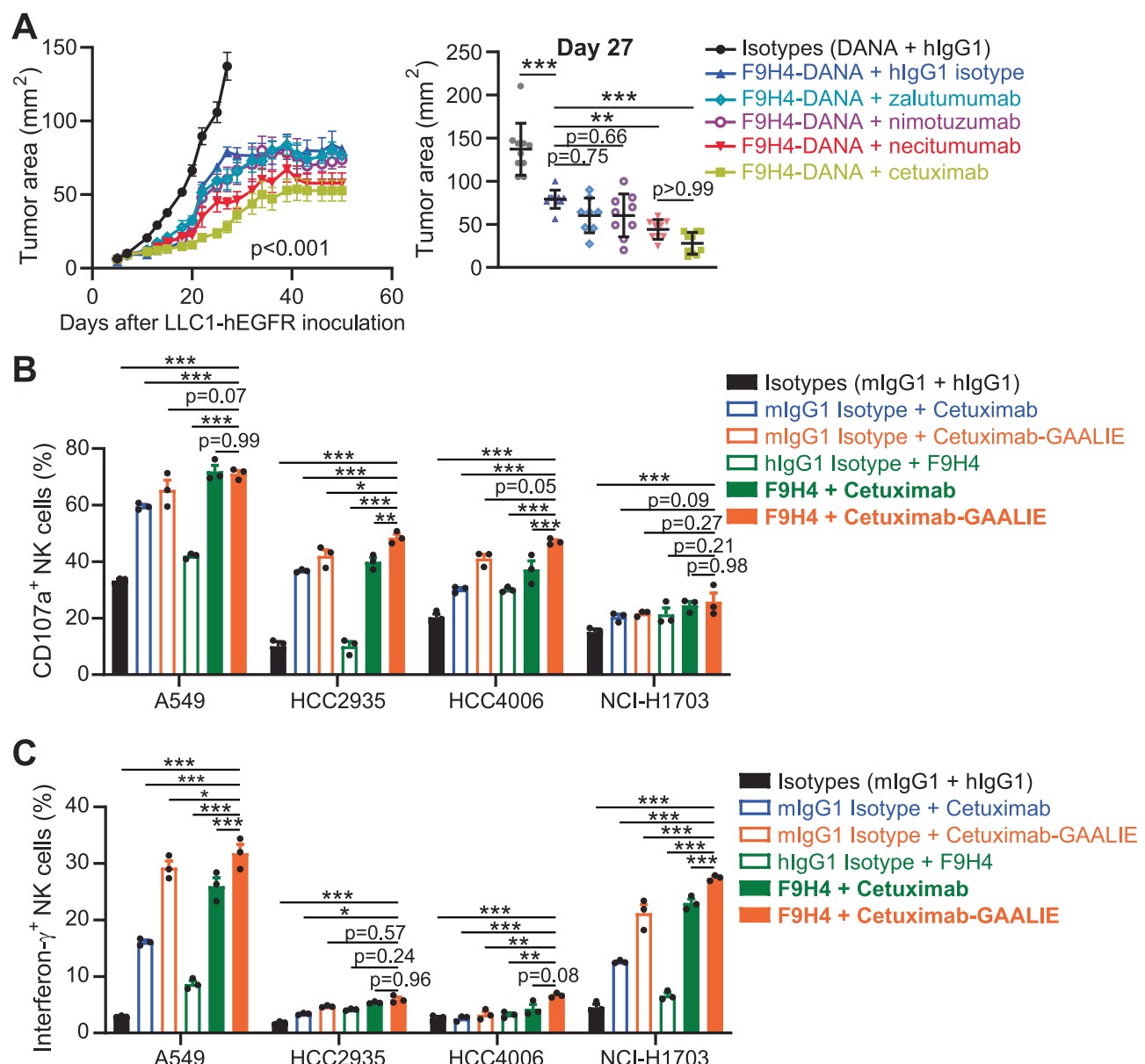

**Fig. 5 | Analyses of the synergism between F9H4 and a panel of anti-EGFR antibodies. A** Male adult hFcR mice were inoculated subcutaneously with $1.5 \times 10^6$ LLC1-hEGFR cells and tumors were measured by digital caliper. Treatments with F9H4-hIgG1-DANA, EGFR antibodies, and isotype controls (0.1 mg/antibody/per mouse) were done on days 5, 6, and once per week. Only the combinations of F9H4 with cetuximab and necitumumab inhibited tumor growth to greater extents compared to F9H4 without EGFR antibodies, as highlighted by data in the right graph (Day 27). Isotypes (DANA + hIgG1) $n = 10$, F9H4-DANA + hIgG1 isotype $n = 10$, F9H4-DANA + zalutumumab $n = 9$, F9H4-DANA + nimotuzumab $n = 10$, F9H4-

DANA + necitumumab $n = 10$, F9H4-DANA + cetuximab $n = 10$. F9H4 synergizes with an Fc-enhanced version of cetuximab (cetuximab-GAALIE) to promote NK cell degranulation (**B**) and interferon-γ production (**C**) against a panel of human lung cancer cell lines. Primary NK cells were co-cultured with the indicated cell lines and in the presence of the indicated antibodies, followed by analyses of CD107a and intracellular interferon-γ by flow cytometry. Data represent three (**B**, **C**) or are pooled of two (**A**) independent experiments, are mean ± standard error of triplicates (**B**, **C**), and were analyzed by two-way ANOVA with Bonferroni's test (**A**) or Dunnett's test (**B**, **C**). *$p < 0.05$, **$p < 0.01$, ***$p < 0.001$.

which is a culture technique with non-dissociated tissues in inserts that enable access to media from apical and basolateral sides[36]. Immuno-histochemistry (IHC) identified CD16a/b⁺ cells, presumably leukocytes, in stromal tissues surrounding and inside tumors as well as in areas with high density of tumor cells (Fig. 6A–C). We applied F9H4 to these tumors and normal lung tissue explants, but did not detect differences in CD16a/b expression levels by IHC (not shown). On the other hand, the tumor tissue explants had supernatants with more soluble CD16a/b molecules than normal lung tissue explants and F9H4 lowered this shedding (Fig. 6D). We also administered cetuximab to additional specimens, and discovered that it increased the amounts of soluble CD16a/b molecules shed in these supernatants and, again, F9H4

inhibited this shedding (Fig. 6E). However, a limitation was that it was not possible to compare cetuximab with an isotype control ± F9H4, which would require four groups total, because these tumor samples were small and could only be separated in half. Hence, these studies formally compared F9H4 vs. isotype. By flow cytometry, we detected an abundance of non-neutrophil myeloid cells (presumably macrophages), whereas NK cells and neutrophils were less abundant in the analyzed tumor samples (Fig. 6F). F9H4 increased the CD16a/b expression levels on the surfaces of non-neutrophil myeloid cells that were in tumors and, for one patient, normal lungs (Fig. 6G). In contrast, the CD16a/b expression levels were variable in NK cells and neutrophils and F9H4 did not consistently cause an impact on them

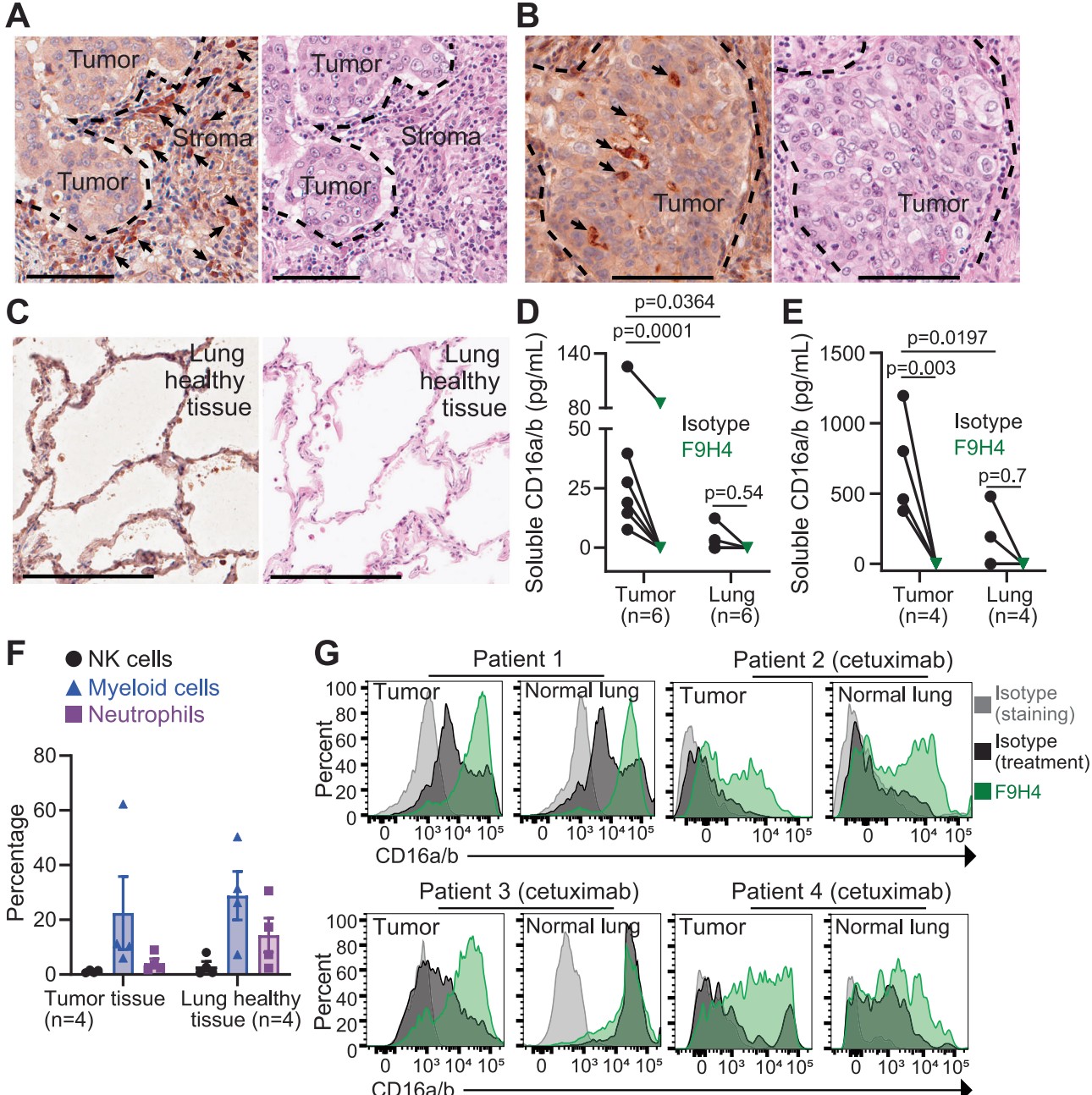

**Fig. 6 | F9H4 inhibits the CD16a/b shedding by tumor-infiltrating leukocytes from untreated patients with early-stage NSCLC. A–C** CD16a/b⁺ cells physically localize in the stromal tissues surrounding and inside tumors (**A**) and areas with high density of tumor cells (**B**). In (**A**, **B**), the dotted lines indicate tumor areas. IHC for CD16a/b and H&E staining of tumor samples (**A**, **B**) and paired normal lung of one patient (**C**). The arrows in **A**, **B** indicate cells that expressed CD16a/b. In **A**, **B**, the scale bars correspond to 100 µm; in **C**, the scale bars correspond to 250 µm. Data are for one patient and represent three patients, thus *n* = 3. F9H4 decreases the shedding of soluble CD16a/b in supernatants of tumor explants that were (**E**) or not (**D**) treated with cetuximab. Tumor explants or paired normal lung tissues were cultured for 4 h with the indicated antibodies, followed by analyses of soluble CD6a/b by sandwich ELISA. **F**, **G** Analyses of tumor and lung-infiltrating myeloid cells, neutrophils, and NK cells. **F** CD11b⁺ myeloid cells are more abundant in tumor and lung tissues than NK cells or neutrophils. **G** F9H4 increased the surface CD16a/b expression levels in these myeloid cells. Data are mean ± standard error (**F**), represent three patients (**A**–**C**), and were analyzed by two-way ANOVA with uncorrected Fisher's LSD test (**D**–**E**).

(Supplementary Fig. 22A, B). Therefore, F9H4 inhibits CD16a/b shedding by tumor infiltrating-myeloid cells from untreated patients with early-stage NSCLC.

## Discussion

We discovered that CD16a/b shedding can be inhibited by a CD16a/b-targeting mAb (F9H4), which as consequence promotes NK cell-mediated cytotoxicity and interferon-γ production against lung cancer in the context of co-administration with two types of EGFR antibodies

(cetuximab and necitumumab). Notably, a previous study showed that CD16a cleavage also occurs in the immune synapse to help NK cells disengage after degranulation and move on to the next target cell, a process termed "serial killing"[37]. However, other previous studies showed that CD16a is internalized after engagement[38,39]. Conceptually, internalized CD16a would not mediate intercellular adhesion. Furthermore, ADAM17 inhibitors, the knocking out of the *ADAM17* gene, and CD16a^{S197P} expression all promoted the NK cell effector functions against cancer cells opsonized by antibodies against tumor surface

antigens[3,5,8,9,16]. Our data did not rule out a potential role of CD16a shedding in serial killing, but they were consistent with the latter studies that showed inhibition of ADCC by CD16a shedding. Furthermore, we found that blood NK cells downregulate CD16a in cetuximab-treated, tumor-bearing mice, and such an effect is presumably by cleavage because F9H4 inhibits it. On the other hand, cetuximab induces surface CD16a downregulation by NK cells that are co-cultured with A549, thus suggesting that CD16a can also be cleaved in the immune synapse. F9H4 inhibits this downregulation and promotes ADCC. Therefore, the aggregate of data and previous studies indicate that CD16a cleavage likely occurs in two contexts that conceptually are mutually exclusive: either before receptor engagement or in the immune synapse. We reason that CD16a cleavage is immunosuppressive when it antecedes receptor engagement.

Tyrosine kinase receptor-blocking antibodies, like cetuximab for EGFR, have a dual mechanism of action: the variable regions block receptor–ligand interactions to induce intrinsic effects on tumor cells, and the Fc binds Fcγ receptors in leukocytes that induce cellular immunity. Consistently, NK cells contribute to the activity of tumor antigen-targeting antibodies through ADCC[2]. Here, we demonstrated that F9H4 promotes ADCC by inhibiting the cleavage-mediated downregulation of CD16a, which is the Fcγ activating receptor in NK cells. Hence, F9H4 represents an opportunity to promote the immune functions of antibodies that, like cetuximab, opsonize tumor cells for elimination by ADCC. However, the infiltration of NK cells is relatively low in human lung cancer and other types of solid tumors[40,41]. F9H4 does not address the often-observed low infiltration of NK cells in the tumor microenvironment; rather, F9H4 overcomes a CD16a cleavage-mediated NK cell dysfunction. Such an approach is particularly relevant in human lung adenocarcinoma because the CD16a expression levels can be lower in tumor-infiltrating NK cells compared to NK cells in adjacent normal lung tissues[42]. Combination therapies may optimize efficacy, such as with cytokine therapies that expand the intratumoral NK cell compartment[43,44]. However, another limitation was that we did not observe CD16a-shedding inhibition by F9H4 in primary tumor-infiltrating NK cells from untreated early-stage NSCLC patients. Since the majority of the CD16a+ cells, presumably leukocytes, were in the stroma, we assume that these tissues represent immune-excluded tumors. Consistent with this idea, cetuximab induces the CD16a/b shedding in tumor explants, likely because it provided leukocyte stimulation through the Fc. Another alternative explanation for why cetuximab did not induce CD16a shedding by these tumor-infiltrating NK cells from patients is that an addition of recombinant interleukin 15 protein may be needed to better support NK cell survival and functions in tumor explants, but in our study we did not add interleukin 15 to the tumor explants. Therefore, we envision that CD16a/b shedding would be particularly relevant in immune-infiltrated tumors.

F9H4 also inhibits CD16b shedding by neutrophils, which, depending on the context, may have beneficial or detrimental roles in the activities of tumor cell-opsonizing antibodies. For example, non-fucosylated anti-CD20 antibodies increase the phagocytosis of lymphoma cells by neutrophils and major histocompatibility complex class II expression by the phagocytic neutrophils[45]. On the other hand, CD16a bispecific antibodies that do not cross-react with CD16b are not removed from the circulation by neutrophils[46]. For these reasons, we speculate that neutrophils would affect the in vivo biological effects of F9H4 in human subjects by phagocytosing tumor cells and/or acting as a "sink" for F9H4 and tumor cell-opsonizing antibodies.

F9H4 provides the opportunity to inhibit, in a highly specific manner, the CD16a/b shedding in vivo. We applied F9H4 to tumor-bearing hFcR mice and discovered that cetuximab induces CD16a shedding by blood NK cells and F9H4 inhibits it. However, cetuximab does not induce CD16a shedding by monocytes, and thereby we did not observe an impact of F9H4 on CD16a expression in blood monocytes in vivo. On the other hand, CD16a expression levels are downregulated in tumor-infiltrating NK cells and macrophages. These contrasting results (blood vs. tumor) suggest that CD16a expression may be regulated by multiple processes in the tumor microenvironment in addition to shedding, such as internalization by receptor engagement and cytokines. Consistent with that notion, blood NK cells expresse more *FCGR3A* mRNA, the gene that encodes CD16a, than tumor-infiltrating NK cells in bladder cancer and that was driven by transforming growth factor receptor β[47].

To increase the impact of F9H4 on the CD16a-driven tumor immunity, we used primarily cetuximab as the enabling agent. However, cetuximab is not a standard therapy for lung cancer because it did not generate consistent clinical benefits in phase II and III trials[48]. Nevertheless, cetuximab remains an important tumor-targeting antibody because EGFR overexpression is common in NSCLC[49]. Furthermore, GAALIE mutations increase the cetuximab's immune properties, which in turn are further enhanced by the CD16a-shedding inhibition. For clarification, GAALIE mutations are known to increase the binding affinities between hIgG1 Fc and the Fcγ activating receptors. Consequently, GAALIE-mutant hIgG1 displays more potent Fc effector functions[33]. Hence, cetuximab could be reassessed as an immunotherapy in lung cancer, this time in combination with F9H4 to promote anti-tumor immune responses. F9H4 also synergizes with necitumumab, which is another EGFR antibody and approved for squamous NSCLC, although it did not generate complete responses in a phase-3 trial[32]. However, such synergism is not observed for zalutumumab or nimotuzumab. The discrepancy in efficacy indicates the potential importance of antibody properties, such as epitope, affinity, or antigen internalization by the EGFR antibodies. Consistent with that notion, our experiments revealed that nimotuzumab was a weaker binder to recombinant EGFR compared to the other three mAbs, and cetuximab better bound to LLC1-hEGFR cells than zalutumumab.

In summary, we developed a pharmacological approach for inhibiting, in a CD16a/b-specific manner, the shedding of CD16a/b from cellular surfaces. F9H4 binds CD16a/b and bypasses the multi-substrate specificity of ADAM17. Since F9H4 does not block CD16a, it promotes NK cell-mediated ADCC and inhibits tumor growth in vivo in the context of co-administration with EGFR antibodies, which served here, in this study, as tumor antigen-targeting antibodies. F9H4 and cetuximab-GAALIE synergize to promote ADCC, thus indicating that the inhibition of CD16a shedding and higher binding affinity between CD16a and Fc are two complementary approaches to maximize NK cell stimulation. Furthermore, F9H4 alone inhibits tumor growth in immunocompetent mice, presumably by promoting the Fc-effector functions of endogenous antibodies against tumor cells. Therefore, antibody-mediated inhibition of CD16a cleavage is a new means to potentiate the immune functions of Fc-enabled antibody therapeutics by promoting NK cell effector functions.

## Methods
### Experimental design
The objective of this study was to develop a new pharmacological means to promote NK-cell-driven immunity against cancer. All the cell lines and mouse strains used in this study were previously described, whereas the primary cells were obtained in a de-identified manner from volunteer donors (New York Blood Center) and lung cancer patients who received the standard-of-care (The Mount Sinai Hospital)−details below. The study received approvals by the Institutional Biosafety Committee (SPROTO202200000114), Institutional Animal Care and Use Committee (IPROTO202100000044 and PROTO201900599), and Internal Review Board (STUDY-12-00145-CR003). In the in vivo experiments, the mice were randomly assigned to the different antibody treatment groups. No data points were omitted from any of the analyses.

## Generation of mouse anti-human CD16a antibodies

The antibodies were generated by hybridoma technology, which was performed by a hired company (Promab Biotechnologies, Richmond, California). For the generation of F9H4, Balb/c mice (The Jackson Laboratory catalog number 000651) were immunized with a chimeric protein consisting of the extracellular region of human CD16a (UniProtKB P08637) linked to a human Fc tag. For the generation of 4H2F3 and 8E3B7, Balb/c mice were immunized with a chimeric protein consisting of the D1 domain of human CD16a linked to human IgG1 Fc tag (details below). The F9H4 hybridoma was sequenced by Promab, which provided the DNA and protein sequences for the generation of recombinant antibodies in our laboratory (details below). For the generation of Fab and Fab2, F9H4 as murine IgG1 (mIgG1) was subjected to enzymatic digestion through Pierce™ Mouse IgG1 Fab and F(ab')2 Preparation Kit (Thermo Fisher Scientific, catalog number 44980) and following the technical produces provided by this manufacturer.

## Cell lines

The A549 (CRM-CCL-185), HCC-2935 (CRL-2869), HCC-4006 (CRL-2871), NCI-H1703 (CRL-5889), SKOV3 (HTB-77), Raji (CCL-86), SU-DLH4 (CRL-2957), 293T (CRL-3216), LLC1 (CRL-1642), and B16F10 (CRL-6475) cell lines were purchased from the American Type Culture Collection (Manassas, Virginia). OCI-Ly10, KPN1.1, MOLM13, and C1498-MICB were described[50–53]. The FreeStyle CHO cell line (catalog number R80007) was purchased from Thermo Fisher Scientific. The A549, HCC-2935, HCC-4006, NCI-H1703, Raji, OCI-Ly10, SU-DLH4, SKOV3, and MOLM13 cell lines were cultured in RPMI-1640 media (Gibco catalog number 11875093) whereas LLC1, KPN1.1, B16F10, C1498-MICB, and 293T were cultured in DMEM media (Gibco catalog number 11965092), with both media being complemented with 10% fetal bovine serum (FBS, Biowest catalog number 024N23), 1× Glutamax (Gibco catalog number 35050061), 1× HEPES (Gibco catalog number 15630080), and 1× penicillin/streptomycin (Gibco catalog number 15140122). The FreeStyle CHO cell line was cultured in FreeStyle CHO Expression Medium (Gibco catalog number 12651022) supplemented with 1× HEPES, 4× Glutamax, and 1× anti-clumping agent (Gibco catalog number 0010057AE). All the cell lines were cultured at 37 °C in 5% $CO_2$, except the FreeStyle CHO cell line that was cultured at 37 °C in 10% $CO_2$. All the cell lines were routinely tested negative for mycoplasma contamination with MycoAlert Mycoplasma Detection Kit (Lonza catalog number LT07-318). A549, LLC1, C1498-MICB, B16F10, and KPN1.1 were also routinely tested negative for the presence of rodent pathogens through and IMPACT PCR Profile (IDEXX BioAnalytics catalog number 41-00021).

## Cell line transduction with lentiviral vectors

For the generation of human EGFR (hEGFR)-expressing LLC1 cells, lentiviruses were produced in 293T cells by transient transfection with pHAGE vector plasmids (Addgene, plasmid #116731), packaging plasmid (pCMV-DR8.9.1), and envelope plasmid (pCMV-VSV-G) through the TransIT-293 Transfection Reagent (Mirus catalog number MIR 2704). Supernatant-containing lentiviruses were added to LLC1 cultures, which then underwent puromycin-mediated selection. The pHAGE vector consists of the hEGFR cDNA followed by IRES and puromycin resistance gene, all under control of the EF1α promoter. The resulting cell line was named as "LLC1-hEGFR". Furthermore, for the generation of the three MOLM13 cell lines (control vector, CD16a WT, and CD16a S197P), the following three pHAGE vector plasmids were generated: (1) control vector, which is the pHAGE vector with the EF1α promoter followed by multiple cloning sites, IRES, and ZsGreen, (2) CD16a WT:, which is a pHAGE vector with the replacement of ZsGreen for the human FCER1G cDNA plus the insertion of human FCGR3A (F176) cDNA in the multiple cloning sites, and (3) CD16a S197P, which is a pHAGE vector with almost the same sequence of the CD16a WT

plasmid, except that the S197P mutation was included in the human FCGR3A cDNA. The lentiviruses with these pHAGE vectors were produced in 293T cells using also pCMV-DR8.9.1 and pCMV-VSV-G. Supernatant-containing lentiviruses were added to MOLM13 cell cultures, and the cells were sorted by flow cytometry using a BD Aria after staining with anti-CD16a antibody (clone 3G8) or based on ZsGreen expression (control vector). On the other hand, LLC1-hEGFR cells were analyzed by flow cytometry after staining with cetuximab or zalutumumab at a concentration range of 0.78–12.5 µg/mL, which was revealed with a secondary antibody against human IgG.

## Primary NK cell, macrophage, and neutrophil isolations and cultures

De-identified buffy coats were purchased from the New York Blood Center (Queens, New York). We did not obtain private information, such as sex and age, from the volunteer donors. Samples were diluted 1:1 in PBS, and the blood mononuclear cell fractions were isolated by Ficoll density gradient centrifugation. On one hand, NK cells were isolated by negative selection using the MojoSort Human NK Cell Isolation Kit (Biolegend catalog number 480054) and cultured in RPMI-1640 media supplemented with 10% FBS, 1× Glutamax, 1× HEPES, 1× penicillin/streptomycin, 5% heat-inactivated human AB serum (Valley Biomedical catalog number HP1022HI), 20 ng/mL recombinant human interleukin 15 (Biolegend catalog number 570306), and 1000 U/mL recombinant human interleukin 2 (Biolegend catalog number 589106). On the other hand, monocytes were isolated by adherence in tissue culture flasks followed by differentiation to macrophages through ≥7-days culture in RPMI-1640 media supplemented with 10% FBS, 1× Glutamax, 1× HEPES, 1× penicillin/streptomycin, and 10 ng/mL granulocyte-macrophage colony-stimulating factor (Biolegend catalog number 572904). Neutrophils were isolated through the lysing of the red blood cells after gradient centrifugation and buffy coat collection; they were cultured overnight using the same media formulation used for monocytes.

## Expression of recombinant proteins

UCOE Hu-P expression vectors (EMD MilliPore catalog number 5.04867.0001) were used for the production of recombinant proteins in the FreeStyle CHO System and 293T cells. For F9H4, the cDNA sequences of the heavy chains and light chains were ligated in the same vector but separated by a viral 2A peptide sequence. The mIgG1 Fc segment of F9H4 was replaced by human IgG1 (hIgG1) with the D265A and N297A (DANA) mutations. FreeStyle CHO cells were transiently transfected with the UCOE vector that enables the expression of the DANA-mutant hIgG1 version of F9H4. For antibody production in bulk, which enabled in vivo experiments, 293T cells were stably transfected with the UCOE vector through puromycin-mediated selection. F9H4-hIgG1-DANA was isolated from supernatants through Protein G sepharose affinity columns (Cytiva catalog number 17061801), followed by elution in pH 2.5, pH neutralization, and buffer exchange to PBS. Short-term storage was at 4 °C and long-term storage at −80 °C. Furthermore, the cDNA sequence of the D1 domain (amino acid position 24-105) of CD16a was ligated with hIgG1 in the UCOE Hu-P expression vector. The D2 domain (amino acid position 107–189) of CD16a was also ligated with hIgG1 in a separate UCOE Hu-P expression vector. The recombinant proteins were produced using the FreeStyle CHO System and isolated in protein G columns. On the other hand, the D1 + D2 protein, which is the full extracellular domain of CD16a linked to a human Fc tag, was produced by ProMab, and that was the same antigen used in mouse immunizations for the generation of F9H4, as explained above.

## Mice

hIL-15 NOG mice (catalog number 13683-M) were purchased from Taconic. Fcγ receptor-humanized mice (hFcR mice) were kindly

provided by Jeffrey Ravetch (The Rockefeller University) and bred in the Center for Comparative Medicine and Surgery (CCMS, Mount Sinai). hFcR mice were described[28]. All mice used in experiments were age-matched and ≥6 weeks old. hIL-15 NOG mice were male. Pilot experiments (not shown) revealed that the LLC1-hEGFR cell line developed larger subcutaneous tumors in male mice than in female mice, and thereby the majority of the experiments were done in male mice only. The mice were kept in the Center for Comparative Medicine and Surgery (Mount Sinai), which is a rodent barrier facility. Experimental mice were co-housed with control mice. $CO_2$ in saturation chambers was the method of euthanasia. The humane endpoints were characterized by the development of any of these following physical signs of disease: moribund state, ≥20% loss of body weight, limb paralysis, curved posture, ulcerating subcutaneous tumors, subcutaneous tumors ≥1 cm diameter as determined by measurement with digital caliper, difficulty breathing, and body condition score ≥2; we adhered to these humane endpoints, by euthanizing mice that developed any of these physical signs of disease. Mice were monitored approximately every day, and tumors were measured with digital calipers for approximately three times per week. The experiments in mice were approved by the Institutional Animal Care and Use Committee in the Icahn School of Medicine at Mount Sinai (New York).

## Enzyme-linked immunosorbent assay (ELISA)

Nunc MaxiSorp ELISA plates (Biolegend catalog number 423501) were coated overnight at 4 °C with 0.1 µg of recombinant human CD16a (Biolegend catalog number 777106), recombinant human CD16b (ACROBiosystems catalog number CDB-H5227), F9H4 (mIgG1), mIgG isotype controls (1, 2a, 2b, and 3, Biolegend catalog number 400102, Bio X Cell catalog numbers BE0085 and BE0086, and Biolegend catalog number 401302, respectively), F9H4-hIgG1-DANA, WT hIgG1 control (Bio X cell, catalog number BE0297), or 7C6-hIgG1-DANA, which is a MICA and MICB antibody. Subsequently, the wells were washed with 1× tris-buffered saline-tween (TBST) buffer and blocked with 2% bovine serum albumin for 2 h. For 3G8 competition assay, the antibodies (F9H4, MOPC21, and 3G8) were co-incubated for 1 h with biotinylated 3G8 (1 µg/mL, Biolegend catalog number 302004). For assessment of the reactivity against CD16a, the antibodies F9H4 (as mIgG1 or as hIgG1-DANA), F9H4 as Fab or Fab2 fragments, 4H2F3, 8E3B7, MOPC21 (mIgG1 isotype control), hIgG1 isotype control, or 7C6-hIgG1-DANA were added at the concentrations indicated in the figures and/or legends. After 2 h, the ELISA plates were washed with TBST, and biotinylated secondary antibodies were added. The biotinylated secondary antibodies were as follows: anti-mIgG1 (RMG1-1, Biolegend catalog number 406604), anti-mouse IgG2a (RMG2a-62, Biolegend catalog number 407104), anti-mouse IgG2b (RMG2b-1, Biolegend catalog number 406704), mouse IgG3 (RMG3-1, Biolegend catalog number 406803), polyclonal anti-mouse IgG (catalog number BAF018, R&D Systems), polyclonal anti-mouse IgG Fab and Fab2 (315-005-006, Jackson ImmunoResearch Laboratories), and anti-human IgG Fc (M1310G05, Biolegend). Furthermore, recombinant human CD64 protein (Biolegend catalog number 790004) was biotinylated using Sulfo-NHS-Biotin (A35358, Thermo Fisher Scientific) and used in Fc receptor binding assays. Peroxidase-conjugated streptavidin (016-030-084, Jackson ImmunoResearch Laboratories), TMB substrate (Biolegend catalog number 421101), and 1 N sulfuric acid were used to reveal and stop the reaction. Absorbance at 450 nm was read in POLARstar Omega microplate reader or PerkinElmer EnVision multilabel reader.

## Cetuximab-GAALIE

The production of this customized antibody was outsourced to IchorBio (England). The binding to human EGFR was validated by ELISA, using the procedures described above but with the coating of multi-well plates with human EGFR recombinant protein (EGR-H5222, ACROBiosystems).

Similarly, for CD16a binding assays, cetuximab-GAALIE, standard cetuximab, or a DANA-mutant human IgG1 molecule were coated in multi-well plates and incubated with biotinylated recombinant human CD16a-V176 and F176 proteins (CDA-H82E9 and CDA-H82E8, ACROBiosystems). The steps with secondary antibodies (for EGFR ELISA), peroxidase-labeled streptavidin, TMB, and sulfuric acid were performed as above. Absorbance at 450 nm was read in POLARstar Omega microplate reader or PerkinElmer EnVision multilabel reader.

## Flow cytometry-based bead assay

BD A4 beads were chemically conjugated to 7C6 as hIgG1 or hIgG1-DANA. The conjugation was achieved by using the Functional Bead Conjugation Buffer Set (558556, BD Biosciences). For analyses of competition, the hIgG1-conjugated beads were incubated for 1 h with the murine version of F9H4, an isotype control (MOPC21), or the commercially available CD16a antibody clone 3G8 (Biolegend catalog number 302002) at the concentrations of 30, 10, 3, and 1 µg/mL in PBS. Subsequently, the beads were incubated for 1 h with 1 µg/mL biotinylated recombinant human CD16a protein (V176, Sino Biological catalog number 10389-H27H1-B), followed by 30-min incubation with phycoerythrin-labeled streptavidin (554061, BD Pharmingen). Of note, there were washing steps with PBS after each incubation. The beads were analyzed in a BD LSRII Fortessa, and the data were analyzed in FlowJo.

## CD16a-engagement reporter assay

CD16a signaling was measured using the Jurkat cell–derived reporter cell line that contains an integrated nuclear factor of activated T-cells (NFAT)-driven firefly luciferase reporter gene, as reported[54]. In this assay, CD16a signaling activates the NFAT transcription factor, inducing expression of firefly luciferase driven by an NFAT responsive promoter. To examine the level of antibody-induced Fc-signaling in the current study, the CD16a reporter cell line was incubated with 10-fold serial dilutions of F9H4, MOPC21, or 3G8 that started from a top concentration of 50 µg/ml for 16 h. After 16 h, the cells were lysed and the levels of firefly luciferase activity were determined using a luciferase assay kit (E1500, Promega). Reporter cells were incubated for the same period in the absence of antibodies to determine the levels of antibody-independent luciferase background production, which were subtracted from the signal to yield the antibody-specific activation in relative light units.

## CD16a-shedding assays by flow cytometry

Whole PBMCs, primary NK cells, or monocyte-derived macrophages were obtained from volunteer donors as explained above. MOLM13 expressing the control vector, CD16a WT, or CD16a S197P were treated for 1–4 h, as indicated in the figure legends, with 5 ng/mL phorbol 12-myristate 13-acetate (PMA, from Apexbio Technology and MedChemExpress, catalog numbers N2060 and HY-18739) or the PMA concentrations indicated in the figures and figure legends. Furthermore, F9H4, F9H4-hIgG1-DANA, 4H2F3, and 8E3B7, MOPC21, F9H4 Fab and Fab2 fragments, hIgG1 isotype, and 7C6-hIgG1-DANA were added simultaneously with PMA and at a concentration range of 0.01–100 µg/mL, as indicated in the figures. After the treatments, NK cells and monocyte-derived macrophages were stained with anti-CD56 (clone HCD56) and CD69 (clone FN50) antibodies and anti-CD14 antibody (clone 63D3), respectively, plus anti-CD16a antibody (clone 3G8) and zombie yellow (423104, Biolegend) and analyzed by flow cytometry, using a BD LSRII Fortessa followed by analysis with FlowJo.

## CD16a-shedding assay by ELISA

The surface proteins of primary NK cells were labeled with biotin using Sulfo-NHS-Biotin and the cells were used immediately after that. These "biotinylated NK cells" were treated for 4 h with 5 ng/mL PMA plus 10 µg/mL F9H4 or MOPC21. Subsequently, the supernatants were

collected and added to ELISA plates that had been coated with anti-CD16a antibody (clone 3G8), which is utilized to capture the biotiny-lated CD16a that is released by cleavage. Recombinant biotinylated human CD16a was used as standard. The capture of biotinylated CD16a (shed by NK cells or the recombinant protein used as control) was revealed by peroxidase-labeled streptavidin, TMB, and 1 N sulfuric acid. Absorbance at 450 nm was read in a POLARstar Omega micro-plate reader.

## CD16b-shedding assay
Whole PBMCs or neutrophils were treated for 1 h with 10 µg/mL PMA plus F9H4 or MOPC21 in concentration range of 0.003–10 µg/mL. Subsequently, the cells were detached using 10 mmol/L EDTA followed by staining with 3G8 for CD16b expression analysis by flow cytometry, using a BD LSRII Fortessa and FlowJo. Neutrophils were identified through the CD15 marker.

## NK cell effector function assays
NK cells were isolated by negative selection from buffy coats and cultured as explained above. For CD107a externalization assays, NK cells were co-cultured with A549, HCC-2935, HCC-4006, NCI-H1703, HCT-11, RKO, Raji, or SKOV3 in 1:1 ratio for 4 h in the presence of 5 µg/mL F9H4 or MOPC21, cetuximab (PSC-24-274, Enzo Life Sciences), cetuximab-GAALIE, rituximab (ENZABS4190200, Enzo Life Sciences), trastuzumab (PSC-24-087, Enzo Life Sciences) or hIgG1 isotype at concentration range of 0.001–3 µg/mL, as indicated, and alexa[488]-labeled anti-human CD107a (H4A3, Biolegend). Furthermore, mon-ensin (420701, Biolegend) was added after the first hour of co-culture. For intracellular interferon-γ assays, NK cells were co-cultured with A549, HCC-2935, HCC-4006, or NCI-H1703 cells for 6 h with 5 µg/mL F9H4 or MOPC21 and cetuximab, cetuximab-GAALIE, or hIgG1 isotype at a concentration range of 0.001–3 mg/mL in the presence of 1× brefeldin-A (420601, Biolegend). Subsequently, these cells were fixed with Biolegend Fixation Buffer (420801), permeabilized with Biole-gend Intracellular Staining Permeabilization Wash Buffer (421002), and stained with anti-human interferon-γ antibody (4S.B3, Biolegend). For the cytotoxicity assays, NK cells and A549 cells were co-cultured for 4 h followed by cell detachment using 10 mmol/L EDTA and label-ing with 7-AAD (420403, Biolegend). In these three assays (CD107a externalization, intracellular interferon-γ, and dead cell analysis) NK cells were identified by the labeling with anti-human CD45 (2D1, Bio-legend) and/or CD56 (HCD56, Biolegend). Cells were analyzed in BD LSRFortessa and data were analyzed in FlowJo.

## NK cell proliferation assay
Blood NK cells were isolated by negative selection from volunteer donors and cultured for 7 days in media supplemented as above. Then, NK cells were labeled with carboxyfluorescein diacetate succinimidyl ester (CFSE) (423801, Biolegend) and cultured for another 3 days in media supplemented as above, plus 10 µg/mL 3G8, F9H4 as mIgG1, or isotype control. NK cells were then analyzed by flow cytometry.

## ADCP assays
Total or intermediate and non-classical monocytes were isolated from human PBMC through adherence or CD16⁺ Monocyte Isolation Kit-Human (Miltenyi Biotec, catalog number 130-091-765), respectively, and induced to differentiate to macrophages through 10 ng/mL granulocyte-macrophage colony-stimulating factor for 7–14 days. On the other hand, A549, Raji, OCI-Ly10, and SU-DLH4 cells were labeled with 10 µmol/L CFSE and co-cultured for 1 h with macrophages and in the presence of 10 µg/mL cetuximab, rituximab, trastuzumab, or human IgG1 isotype control plus F9H4 as murine IgG1 or murine IgG1 isotype control. Cells were then detached by 10 mmol/L EDTA plus scratching, stained with anti-human CD45 and CD11b antibodies, and analyzed by flow cytometry.

## Human lung cancer model in vivo
hIL-15 NOG mice were inoculated intravenously with $1 \times 10^6$ human NK cells, which were freshly isolated by negative selection from buffy coats of volunteer donors (New York Blood Center). Seven days later, the mice were inoculated intravenously with $1.5 \times 10^6$ A549 cells. The antibody treatments were administered to mice by intraperitoneal injections at the doses of 0.1 mg per antibody and on days 1, 3, 7, and 14 relative to tumor cell inoculation. The antibodies used were F9H4 (as mIgG1), MOPC21, cetuximab (as hIgG1, Bio X cell catalog number SIM0002), and hIgG1 isotype control (Bio X cell, catalog number BE0297). In separate experiments, NK cells were inoculated on day 10 after A549 inoculation. The mice were euthanized by $CO_2$ on day 21 after tumor cell inoculation. The lungs were analyzed by histopathol-ogy, which consisted of paraffin embedding and hematoxylin-eosin staining in the Biorepository and Pathology Core (Mount Sinai). Slides were digitalized in Hamamatsu NanoZoomer S210 Digital Slide Scan-ner, and the metastases were manually counted using the Hamamatsu NDP.view2Plus software. Separate cohorts of hIL-15 NOG mice under-went validation of NK cell reconstitution two weeks after NK cell inoculation. To do so, mice were euthanized by $CO_2$, blood was col-lected, and NK cells were analyzed by flow cytometry with the fol-lowing markers: anti-mouse CD45.1 (A20), anti-human CD45 (2D1), anti-human CD56 (HCD56), anti-human CD16a (3G8), and Zombie Yellow, all from Biolegend. Samples were acquired in a BD LSRII For-tessa and analyses were done in FlowJo software.

## The LLC1-hEGFR and LLC1-MICA subcutaneous tumor models in hFcR mice
Male, ≥6 week old hFcR mice were shaved with a groomer and inocu-lated subcutaneously with $1.5 \times 10^6$ LLC1-hEGFR or LLC1-MICA cells. On days 5 and 6, and once per week, the mice were treated with 0.1 mg of each one of the antibodies by intraperitoneal injections. The antibodies used were F9H4, F9H4-hIgG1-DANA, hIgG1-DANA isotype control, MOPC21, hIgG1 isotype control, 7C6-hIgG1, cetuximab, zalatumumab (ICH5116, IchorBio), nimotuzumab (ICH4008, IchorBio), and necitumu-mab (ICH5121, IchorBio). Tumor areas were measured by a digital caliper. The humane end-point was characterized by a moribund state, body condition score ≥2, body weight loss ≥20%, tumor diameter ≥1 cm, and/or ulcerating tumors. The mice were euthanized by $CO_2$.

## NK cell depletion experiments
hFcR mice under the LLC1-hEGFR model were treated by intraper-itoneal injections with 0.1 mg of the following antibodies on days −1 and 0, and once per week relative to tumor cell inoculation: C1184 (isotype control) and anti-NK1.1 antibody clone PK136, both from Bio X Cell (BE0036).

## Macrophage depletion experiments
hFcR mice under the LLC1-hEGFR model were treated by intratumoral injections with 100 µL clodronate or control liposomes (CLD-8901, Encapsula NanoSciences) on days 5, 6, and once per week relative to tumor cell inoculation.

## Analyses of blood and tumor-infiltrating NK cells in the LLC1-hEGFR model in hFcR mice
hFcR mice were subjected to the LLC1-hEGFR model and antibody treatments as described above. On day 21, the mice were euthanized by $CO_2$ and tumors were collected. Blood samples were also collected, by cardiac puncture after euthanasia. Tumors were enzymatically dis-sociated with 1 mg/mL type IV collagenase, 0.1 mg/mL hyaluronidase, and 20 units/mL DNase, whereas blood had the erythrocytes lysed with ACK buffer (A1049201, Gibco). Cells suspensions were stained with anti-mouse CD45.2, CD3ε, NK1.1, and CD49b antibodies, anti-human CD16a/b antibody (clone 3G8) and Zombie Yellow, acquired in a BD LSRII Fortessa, and analyzed in FlowJo.

## Tumor and leukemia models in hFcR mice

hFcR mice (female, 6–8 weeks old) were inoculated intravenously with $5 \times 10^5$ B16F10 cells or $3 \times 10^6$ C1498-MICB cells, or inoculated subcutaneously with $1.5 \times 10^6$ LLC1 or KPN1.1 cells. Naïve mice, which were not inoculated, were used a control. When these LLC1 or KPN1.1-inoculated mice reach the humane endpoint, they were euthanized by $CO_2$ and blood were collected. On the other hand, C1498-MICB or B16F10-inoculated mice were euthanized by $CO_2$, which was followed by blood collection, on days 21 and 14, respectively. Soluble CD16a/b molecules were analyzed by sandwich ELISA (details below) in serum samples.

## B16F10-MICA metastasis model

hFcR mice (female, 6–8 weeks old) were inoculated intravenously with $3 \times 10^5$ B16F10-MICA cells. On days 1, 2, and 7, the mice were treated with 0.2 mg 7C6-hIgG1 or hIgG1 isotype control plus F9H4-mIgG1 plus mIgG1 isotype control by intraperitoneal injections. Euthanasia on day 14 by $CO_2$ was followed by analyses of metastases in the lungs by stereomicroscopy and blood NK cells by flow cytometry.

## Patient-derived normal lung and tumor explants

Tumor and paired normal lung tissues were from untreated patients with early-stage non-small cell lung patients who underwent surgery as standard-of-care in the Mount Sinai Hospital (New York). These specimens were used fresh. They were cut in half a cultured in Millicell Cell Culture Inserts in RPMI-1640 complemented with 10% FBS, 1× Glutamax, 1× HEPES, 1× penicillin/streptomycin, for 4 h. F9H4 or isotype control ± cetuximab were administered in the beginning of these cultures and at doses of 10 µg/mL each. The supernatants were collected and analyzed by sandwich ELISA. This ELISA uses the anti-CD16a/b antibody clone DJ130c to capture and 3G8 to detect CD16a/b. IHC for CD16a/b expression used a rabbit anti-human CD16a/b polyclonal antibody (Proteintech, catalog #16559-1-AP). IHC and H&E staining were performed in the Biorepository and Pathology Core (Mount Sinai). For flow cytometry analyses, the tumor and normal lung tissues were enzymatically dissociated with 1 mg/mL type IV collagenase, 0.1 mg/mL hyaluronidase, and 20 units/mL DNase, followed by staining with anti-human CD45, CD56, CD3, CD15, CD11b, CD33, and CD64 and Zombie Yellow. Samples were acquired in a BD LSRII Fortessa and analyzed by Flowjo. The collection, storage, and distribution of these patient-derived materials by the Biorepository and Pathology Core (Mount Sinai) were approved by the Institutional Review Board (Mount Sinai) (STUDY-12-00145-CR003), and the use of these specimens in the experiments was approved by the Lung Cancer Tissue Utilization Committee (Mount Sinai). The collected tissues were in excess of what was needed for diagnostic purposes, and they were provided to this study with only a limited dataset.

## Commercial antibodies

This is a list of commercially available antibodies used in this study:
- Anti-human CD11b, Biolegend 101210 (RRID:AB_312793).
- Anti-human CD14, Biolegend 367120 (RRID:AB_2572099).
- Anti-human CD15, Biolegend 323012 (RRID:AB_756018) and 301924 (RRID:AB_2783155).
- Anti-human CD16, Biolegend 302056 (RRID:AB_2564139), 302002 (RRID:AB_314202), 302004 (RRID:AB_314203), Thermo Fisher Scientific PA580622 (RRID:AB_2787918).
- Anti-human CD20, Enzo Life Sciences ENZABS4190200.
- Anti-human CD3, Biolegend 317344 (RRID:AB_2565849).
- Anti-mouse CD3ε, Biolegend 100216 (RRID:AB_493697).
- Anti-human CD33, Biolegend 366623 (RRID:AB_2721556).
- Anti-human CD45, Biolegend 368518 (RRID:AB_2616704), 368522 (RRID:AB_2687375), and 368518 (RRID:AB_2616705).
- Anti-mouse CD45.1, Biolegend 110730 (RRID:AB_1134168).
- Anti-mouse CD45.2, Biolegend 109814 (109814).
- Anti-mouse CD49b, Biolegend 103518 (RRID:AB_2566103)

- Anti-human CD56, Biolegend 318314 (RRID:AB_604103) and 318328 (RRID:AB_10900228).
- Anti-human CD64, Biolegend 399506 (RRID:AB_2861008).
- Anti-human CD69, Biolegend 310916 (RRID:AB_528869).
- Anti-human CD107a, Biolegend 328610 (RRID:AB_1227505).
- Anti-human EGFR, Enzo Life Sciences ENZABS4860200, Bio X Cell SIM0002 (RRID:AB_2894723), IchorBio ICH5116 (RRID:AB_3075761), ICH4008 (RRID:AB_2921528), ICH5121 (RRID:AB_3075766), ICH4004Fc (RRID:AB_3075696).
- Anti-human IgG Fc, Biolegend 410711 (RRID:AB_2565789).
- Anti-human HER2, Enzo Life Sciences ENZABS4900200.
- Anti-human interferon-gamma, Biolegend 502509 (RRID:AB_315235).
- Anti-mouse NK1.1, Bio X Cell BE0036 (RRID:AB_1107737) and Biolegend 108749 (RRID:AB_2564304).
- Human IgG1 isotype control, Bio X Cell BE0297 (RRID:AB_2687817).
- Anti-human IgG1 Fc, Biolegend 410718 (RRID:AB_2721499).
- Mouse IgG1 isotype control, Bio X Cell BE0083 (RRID:AB_1107784) and Biolegend 400197 (RRID:AB_2927801).
- Mouse IgG2a isotype control, Bio X Cell BE0085 (RRID:AB_1107771).
- Mouse IgG2b isotype control, Biolegend 400369.
- Mouse IgG3 isotype control, Biolegend 401301 (RRID:AB_2889135).
- Anti-mouse IgG1, Biolegend 406604 (RRID:AB_315063).
- Anti-mouse IgG2a, Biolegend 407103 (RRID:AB_345323).
- Anti-mouse IgG2b, Biolegend 406704 (RRID:AB_315067).
- Anti-mouse IgG3, Biolegend 406803 (RRID:AB_315070).
- Anti-mouse IgG, R&D Systems BAF018 (RRID:AB_562589).
- Anti-mouse IgG, Fab and Fab2 specific, Jackson ImmunoResearch Laboratories 315-005-006 (RRID:AB_2340034).

## Statistical analyses

All graphs were built with and the statistical analyses were performed in the GraphPad Prism software. All graphs show mean ± standard deviation or standard error, as indicated in the figure legends. The tests used were nonlinear regression, two-tailed unpaired Student's $t$ test, two-tailed Mann–Whitney test, two-way analysis of variance with Bonferroni's test or uncorrected Fisher's LSD test, or Rank-Based Multiple Test Procedures and Simultaneous Confidence Intervals, as indicated in the figure legends. Statistical significances were considered when $p < 0.05$. The in vivo data consist of pooled data from at least two independent experiments, whereas the in vitro data represent at least three independent experiments (further details in the figure legends), except for the in vitro experiment shown in main Fig. 3C that was done twice. The in vitro experiments with cells had at least three wells (triplicates) per condition (technical triplicates), whereas the graphs for the in vivo experiments consist of one mouse per symbol (biological replicates).

## Reporting summary

Further information on research design is available in the Nature Portfolio Reporting Summary linked to this article.

## Data availability

All data are included in the Supplementary Information or available from the authors, as are unique reagents used in this Article. The raw numbers for charts and graphs are available in the Source data file whenever possible. Source data are provided with this paper.

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

## Acknowledgements

We thank Jeffrey Ravetch (The Rockefeller University) for providing the Fcγ receptor-humanized mice, Thomas Pabst (University of Bern) for providing the parental MOLM13 cell line, Sergio Lira (Mount Sinai) for reading the manuscript and helpful discussions, and Xiaoyu Song (Mount Sinai) for statistical support during early conceptualization phases. We thank for the following funding sources to LFdA: National Institutes of Health grant 5R37CA269982, Leukemia and Lymphoma Society grant 6647-23, Department of Defense grants CA210940 and BC241026P1. This study was directly funded by the American Lung Association (grant LCD-1022356), American Cancer Society (grant RSG-24-1188643-01-IBCD), Elsa U. Pardee Foundation, and TCI Developmental Funds Award (National Cancer Institute grant P30CA196521), all to LFdA and the latter also to TUM. This research was conducted with the support of the Biorepository and Pathology Core at Icahn School of Medicine at Mount Sinai. Services and products in support of this research project were generated by the Tisch Cancer Institute Biostatistics Shared Resource, supported, in part, with funding from NIH-NCI Cancer Center Support Grant P30 CA0196521.

## Author contributions

Conceptualization: B.T.d.S.B. and L.F.d.A. Investigation and analyses: B.T.d.S.B., S.Q., S.M., X.Z., R.C.A.P., L.R.d.L.B., M.H., M.F., P.H.A.d.S., R.A., B.K.C., M.A.D., B.H., R.M.F., R.B., T.U.M., and L.F.d.A. Funding acquisition: T.U.M., and L.F.d.A. Project administration: L.F.d.A. Supervision: L.F.d.A. Writing—original draft: L.F.d.A. Writing—review & editing: B.T.d.S.B., R.A., T.U.M., and L.F.d.A.

## Competing interests

L.F.d.A. and B.T.d.S.B. are named inventors in a patent that claimed F9H4 (WO2024026374A1) and filed by Mount Sinai Innovation Partners. L.F.d.A. is also named inventor in two patents that are related to 7C6 (WO2018217688A1 and WO2021072211A1) and filed through the Dana-Farber Cancer Institute; L.F.d.A. received royalty payments in relation to the antibody 7C6. L.F.d.A. also served as consultant in a translational advisory board for Cullinan MICA and has a consultancy agreement with Ono Pharmaceutical. All the other authors had no competing interests.

## Additional information

¹Marc and Jennifer Lipschultz Precision Immunology Institute, Icahn School of Medicine at Mount Sinai, New York, NY, USA. ²Division of Infectious Disease, Department of Medicine, Icahn School of Medicine at Mount Sinai, New York, NY, USA. ³Department of Population Science and Public Policy, Icahn School of Medicine at Mount Sinai, New York, NY, USA. ⁴Department of Thoracic Surgery, Icahn School of Medicine at Mount Sinai, New York, NY, USA. ⁵Department of Pathology, Molecular, and Cell-based Medicine, Icahn School of Medicine at Mount Sinai, New York, NY, USA. ⁶Tisch Cancer Institute, Icahn School of Medicine at Mount Sinai, New York, NY, USA. ⁷Department of Immunology and Immunotherapy, Icahn School of Medicine at Mount Sinai, New York, NY, USA. ⁸Department of Oncological Sciences, Icahn School of Medicine at Mount Sinai, New York, NY, USA. ⁹Present address: Grossman School of Medicine, New York University, New York, NY, USA. ¹⁰Present address: Yale School of Medicine, New Haven, CT, USA. ¹¹Present address: Ichor Biologics, LLC, New York, NY, USA. ✉e-mail: Lucas.FerrarideAndrade@mssm.edu

