## [Transparent Peer Review file · Nature Communications]

A monoclonal antibody that inhibits the shedding of CD16a and CD16b and promotes antibody-dependent cellular cytotoxicity against tumors

Corresponding Author: Dr Lucas Ferrari de Andrade

Version 0:

Reviewer comments:

Reviewer #1

(Remarks to the Author)

This manuscript reports the functionality of a new antibody, named F9H4, isolated from hybridoma technology. The noteworthy result is that this antibody binds CD16a outside of the ligand binding domain, prevents shedding and thereby enhances NK-mediated ADCC. In general, the results are clearly presented and well discussed. However, the introduction could use some restructuring. It is mentioned that CD16b is very homologous and expressed on neutrophils in the first paragraph, but the relevance is unclear until the end with still little clarification. Also the result section could use some restructuring/ additional data. Since this is the first report of a new antibody, it should first make clear how these were generated, i.e. immunizing mice with extracellular CD16a. Just mentioning 'by hybridoma technology' is not sufficient/important. Next, it should be made clear how specific this new antibody is, no crossreactivity with other Fc receptors like CD32a or b? Other comments:

Major points:

- Recombinant proteins of CD16a were produced to test F9H4 binding. The antibody binds to full length CD16a and to D1. Elaborate on why the binding pattern changes to an S-curve and does not plateau like it does to full length CD16.
- Does the DANA mutation affect Fc binding to FcRn and thus half-life? These DANA mutations are not very commonly used like LALAPG, please argue better why these were chosen and use proper references.
- In figure 3C, the authors show a flowcytometry-based tumor killing assay with 7-AAD. 7-AAD was added after the 4 hour incubation of NK+A549 cells, and dead A549 cells without NK cells present were subtracted. Hence, the overall percentage killing then includes both dying NK cells and A549 cells. How is this then a true measure of A549 killing? Why not use a more robust tumor killing assay with luciferase or radioactive chromium? Why not use CFSE to specifically stain target cells, allowing live and dead A549 cells to be quantitatively distinguished. Was a positive control taken along for 100% tumor killing?

Additionally, from text, figure and legend it is not clear that there was gating on NK cells, hence show gating strategy for the NK cell effector function assays in supplemental and elaborate on gating in figure legend.

The author uses the phrase 'cetuximab-tagged tumor cells'. This is not a very immunological statement. Immunologists would say cetuximab-opsonized tumor cells. In general, the manuscript would benefit from a critical review by an immunologist.

- Test other tumor types in tumor killing assays (luciferase or chromium based) to examine extent of effectiveness. Especially since it only worked with half of the EGFR antibodies tested. Does it work better in solid tumors where there is more NK cell infiltration like, e.g. low-stage, high-grade?
- It has been reported that different anti-CD16 antibodies (e.g. CD16, 3G8, B73.1 and MEM-154) influence NK cell expansion rates (Kim et al, 2023, Scientific reports). Does F9H4 affect NK cell expansion?
- In figure 4F, there is a strong significant difference between Isotype+NK1.1 and antibodies+NK1.1, suggesting that the antibodies also exhibit non-NK-cell-dependent anti-tumor activity. Please elaborate.
- The authors introduce a GAALIE mutation to enhance the Fc tail of cetuximab. However, there are papers stating that this mutation reduces binding affinity to all FcγR receptors compared to wildtype (Patel et al, J Immunol, 2023) and that ADCC is impaired with GAALIE Fc mutations. However, glycosylation patterns are important for the enhanced binding. Elaborate more on this in discussion.

- In the discussion, “Therefore we reason that the CD16a cleavage occurs not only in the immune synapse and . . . , page 13, lines 24-25” Please elaborate more, as data suggest that shedding happens in immune synapse after cetuximab treatment.
- Discussion, “However, such synergism was not observed for zalutumumab or nimotuzumab”. Please reason why this is not the case.

Please discuss how binding to CD16b could potentially act as a sink when used in human patients. The mouse models lack this receptor since there are no human PMN present.

Minor points

- On page 6, line 24, it is stated that F9H4 is an agonist, but in line 19 on the same page it is stated that F9H4 is a non-agonist.
- Page 11, lines 9-10 do not read well
- Figure 6G and S14, color legend is missing
-
- The manuscript is missing an abbreviations list

Reviewer #2

(Remarks to the Author)

The manuscript by Bortoleti et al. introduces a novel pharmacological approach to inhibiting CD16 shedding, thereby enhancing anti-tumor activity. The authors describe the development of F9H4, an engineered antibody that binds CD16a/b via its Fab fragment and allosterically prevents CD16 cleavage. They demonstrate that F9H4, when combined with Cetuximab, promotes antibody-dependent cellular cytotoxicity (ADCC) both in vivo and in tissue samples from lung cancer patients. These findings are innovative, clinically relevant, and thoroughly discussed, including the limitations of the study. While the results are generally compelling, the following points should be addressed:

1. Statistical analysis concerns: For example, in Figure 1B, the authors employed ANOVA, which assumes a normal distribution. However, there is no evidence to support this assumption, and the number of observations appears insufficient. A non-parametric alternative, such as the Kruskal-Wallis test, would be more appropriate. Furthermore, for some flow cytometry results, the normal distribution of CD16 expression is clearly violated (e.g., Figure S1, influencing Figures 1C, 1D, and Figure S2, influencing Figure 1F). As such, statistical reassessment is warranted in several figures, including Figures 1E and 1H.

Additionally, in Figures 3A-D, normal distribution cannot be assumed, making three replicates insufficient. The statistical test used for Figures 3A and 3B is not mentioned. Some figures also lack statistical significance markers, making it challenging to interpret or support the authors' claims. The authors should revisit their statistical analyses across the manuscript and update their methodology where needed.

2. Inconsistency in measurement variables: In Figures 1F and 1G, the authors performed similar experiments on two different cell types but presented different metrics: percentage of CD16-expressing cells in Figure 1F and MFI in Figure 1G. A similar discrepancy exists in Figure 1E. The authors should clarify why they chose to present different variables for comparable experiments to ensure consistency and transparency.

3. Negative CD16 values in Figure 1B: The ELISA results in Figure 1B show CD16 levels below zero, which is unexpected for this assay. Potential causes include over-subtracted background, sample preparation errors, or extrapolation of values below the detection threshold of the standard curve. The authors should clarify their methodology and explain these negative values to ensure accurate data interpretation.

4. CD16 internalization: While the authors assessed CD16 expression on the cell surface and in the medium, they hypothesize potential internalization of CD16. To validate this hypothesis, the inclusion of internalization assays, such as intracellular staining, would be beneficial. These experiments could measure CD16 levels following F9H4 treatment, with and without tumor cell challenge, and provide mechanistic insights into cleavage-independent regulation of CD16.

5. Subpopulation analysis: CD16 expression differs between NK cell subpopulations (CD56Dim/CD56Bright). Evaluating the effects of F9H4 on these distinct subpopulations would provide a more comprehensive understanding of its impact on NK cell function.

6. Fresh vs. activated NK cells: The in vitro assays utilized activated NK cells, as stated in the Methods section. However, receptor expression differs between fresh and activated NK cells. Assessing the impact of F9H4 on fresh NK cells would provide additional insights. Specifically, this analysis might help explain the unexpected results in Figures 4D and 4E, where circulating NK cells (likely fresh) seem affected by F9H4, whereas tumor-infiltrating NK cells (likely activated) show no reduction in CD16 expression—or even higher levels—in the absence of the antibody.

7. Figure 4D-H: Alignment of results: As mentioned in the previous section, Figure 4E indicates that CD16 expression on tumor-infiltrating NK cells and macrophages is not affected by F9H4 or Cetuximab. This observation appears to contradict earlier results, where CD16 was significantly modulated by the antibody. The authors should discuss and explain this inconsistency, providing insights into the biological or experimental reasons that might underlie this discrepancy.

8. NK cell validation: The authors should clarify how they validate NK cell integrity after isolation and during the experiments. While the Methods section describes NK cells as CD45+/CD56+ for identification, some figures (e.g., Figure 3) suggest NK cells are considered CD45+ alone. Providing the gating strategy used in the in vitro experiments would enhance clarity and reproducibility.

Reviewer #3

(Remarks to the Author)

To the Authors

In their manuscript entitled "Discovery of a monoclonal antibody that binds CD16a to inhibit the shedding and promote anti-tumor immunity" Bortoleti et al describe the generation and functional characterization of a novel human FcγRIII (CD16 antibody), named F9H4. This FcγRIII directed antibody is shown to inhibit shedding of the receptor and to enhance ADCC by human NK cells. In vivo, co-administration of F9H4 enhanced the efficacy of two (cetuximab and necitumumab) but not of two other (zalutumumab and nimotuzumab) EGFR antibodies in a mouse model of human EGFR transfected LLC1 syngeneic tumor cells in hFcR mice.

Overall, this is an interesting manuscript that describes an approach to inhibit the well-described shedding of human FcγRIII to enhance the efficacy of EGFR antibodies. However, many important issues are not properly addressed

Major:

1. The title claims that the CD16 antibody "promotes anti-tumor immunity", which would suggest a tumor-directed adaptive immune response. This, however, is not shown and cannot be concluded from the presented data. The term "immunity" is often used too loosely (e.g. page 2, line 23; page 7, line 25; page 13, line 13).

2. Throughout the manuscript, the authors emphasize the role of NK cells and ADCC for the therapeutic efficacy of EGFR antibodies in patients. However, potential contributions from other FcγRIII expressing cells are widely neglected. For example, macrophages mediate ADCP and PMN ADCC via EGFR antibodies. The impact of F9H4 on these effector mechanisms could have easily been addressed, at least in vitro.

3. Only EGFR antibodies were used to demonstrate the impact of F9H4. Seeing that the approach would also work with other tumor targeting antibodies would greatly increase the impact of the current manuscript.

4. No explanation is provided why some EGFR antibodies benefit from co-administration of F9H4, while others do not. For example, cetuximab and zalutumumab are very similar EGFR antibodies.

5. In Figure 1, PMA is used to trigger shedding of FcγRIII, but the presentation of data is difficult to understand. Apparently, effector cells (NK cells C + D, macrophages in F and PMN in G) were stimulated with PMA and CD16 expression was measured. Why are results presented as "% expression" in C and F, but as "MFI" in G? Apparently, CD16 shedding was prevented in NK cells, but in macrophages and PMN CD16 expression was raised by F9H4? How is staining with F9H4 affected by shedding of the receptor? Here, representative immunofluorescence blots would be informative. Apparently, the Legends in C and D are mixed up. In I, CD16 expression goes up upon incubation with F9H4-DANA? How were percentages calculated ?

6. Several of the sub-figures (e.g. Figure 1A and E, Figure 2D and H) do not show SEM although apparently more than one experiment was performed. The term "triplicates" should be specified whether technical or biological triplicates are shown.

Minor:

1. The clarity of the Figures (in particular Figure 1) could be improved by adding small cartoons describing the set-up of some of the more complex assays. Also, a graphical abstract of the manuscript would be helpful.

2. Throughout the manuscript, the antibody 3G8 is often described as an anti-CD16a antibody (e.g. line 15-16), which is not correct. 3G8 is pan-CD16, recognizing both CD16a and CD16b. Also the title is misleading since F9H4 is pan-CD16 and not CD16a specific.

3. Apparently F9H4 binds to the D1 domain of FcγRIIIa (Figure 2) and inhibits cleavage, which is supposed to occur between D1 and D2. It would be interesting to model the F9H4 binding site to FcγRIII – similar to Figure 1A. Do other commonly available CD16 antibodies also inhibit shedding?

4. On page 11, line 9 other EGFR antibodies (nimotuzumab, necitumumab zalutumumab) are described as "biosimilars" of cetuximab, which is not correct.

5. Numerous parts of the manuscript require editorial work. E.g.

a. Page 2 line 11: "specificity to hundred types of proteins".

b. Page 2 lines 18 – 21.

c. Suppl. Page 6, line 18: "staining with 3G8 for CD16a expression". PMN express CD16b.

6. Often the Legend of Suppl. Figures state that "data represent three independent experiments". Here, "means +/- SEM" should be depicted to give the reader an impression about the variation of results.

Version 1:

Reviewer comments:

Reviewer #1

(Remarks to the Author)

The manuscript has significantly improved in the revision.

Reviewer #2

(Remarks to the Author)

No further comments. The authors responded to all criticisms

Reviewer #3

(Remarks to the Author)

The authors provide an extensively revised manuscript, in which important biological questions are still not properly addressed:

1. Why is NK cell mediated ADCC by cetuximab enhanced, while that by zalutumumab is not affected? Both EGFR antibodies bind to overlapping epitopes of EGFR and mediate similar levels of ADCC. If inhibition of FcγRIII shedding is the relevant mechanism of enhanced ADCC by F9H4 why is it so highly dependent on the EGFR fine epitope? How about additional EGFR antibodies ?
2. Why did the authors not use antibodies against other target antigens like HER-2/neu or CD20 to demonstrate that F9H4 does not only work with cetuximab?
3. With macrophages as effector cells, apparently only A549 cells were tested as target cells in vitro. In the published literature other tumor cell lines with higher EGFR expression levels were shown to be phagocytosed by macrophages in the presence of EGFR antibodies. Again, antibodies against other target antigens (please see also above) could be used to demonstrate broader applicability of this novel approach.

Version 2:

Reviewer comments:

Reviewer #3

(Remarks to the Author)

Dear Authors,

thank you very much for your efforts to address my concerns and to broaden the applicability of your findings by testing additional antibodies and target cell lines.

Overall, I am not impressed by the new results and still wonder what is special about EGFR and cetuximab. To my understanding inhibition of FcγRIIIa shedding should have stronger effects also with other antibodies and additional cell lines.

Authors' comments:

Dear Reviewers, thank you for carefully revising our manuscript. Overall, we felt that the critiques were positive and contributed to improve the scientific quality of our manuscript. We appreciate the reviewers' time for each critical point highlighted in the manuscript and several encouraging remarks such as "...*the results are clearly presented and well discussed*" by Reviewer 1, "*These findings are innovative, clinically relevant,...*" by Reviewer 2, and "*Overall, this is an interesting manuscript...*" by Reviewer 3. We thoroughly revised the manuscript to answer the Reviewers' concerns and with additional data to elucidate all the questions raised. In summary, we included new *in vivo* data in Figure 4I and several new supplementary figures, applied nonparametric tests as requested, and elaborated on the Introduction and Discussion as suggested. Below we provide detailed responses to the Reviewers' concerns.

Reviewer's Comments:

Reviewer #1 (Remarks to the Author)

This manuscript reports the functionality of a new antibody, named F9H4, isolated from hybridoma technology. The noteworthy result is that this antibody binds CD16a outside of the ligand binding domain, prevents shedding and thereby enhances NK-mediated ADCC. In general, the results are clearly presented and well discussed. However, the introduction could use some restructuring. It is mentioned that CD16b is very homologous and expressed on neutrophils in the first paragraph, but the relevance is unclear until the end with still little clarification. Also the result section could use some restructuring/ additional data. Since this is the first report of a new antibody, it should first make clear how these were generated, i.e. immunizing mice with extracellular CD16a. Just mentioning 'by hybridoma technology' is not sufficient/important. Next, it should be made clear how specific this new antibody is, no crossreactivity with other Fc receptors like CD32a or b? Other comments:

Dear reviewer, as requested, we performed additional ELISAs to demonstrate specificity. F9H4 bound CD16a but it did not bind recombinant human CD32a, CD32b/c, or CD64 proteins (Figure S8C). We also included at the beginning of the Results the information that F9H4 was generated by immunizing Balbc mice with human CD16a full-length extracellular domain. The Methodology, in the Supplementary Materials, has additional information. Furthermore, we restructured and added information to the Introduction to enhance clarity by describing CD16a and CD16b in macrophages and neutrophils, respectively, in a separate paragraph (manuscript page 4, lines 16-21).

Major points:

- Recombinant proteins of CD16a were produced to test F9H4 binding. The antibody binds to full length CD16a and to D1. Elaborate on why the binding pattern changes to an S-curve and does not plateau like it does to full length CD16.

Our interpretation is that most, but not all of the F9H4 epitopes are in D1. Hence, the binding looks weaker when only D1 is present. Consistent with that concept, immunizing mice with D1 did not generate mAbs that inhibit the CD16a shedding. In comparison, we did not detect robust binding of F9H4 to D2. We included that information in the Results (manuscript page 7, lines 10-12).

- Does the DANA mutation affect Fc binding to FcRn and thus half-life? These DANA mutations are not very commonly used like LALAPG, please argue better why these were chosen and use proper references.

We attempted to demonstrate the antibody binding to FcRn with a biochemical assay, as we did for CD64 (Figure S8B). However, we did not detect any binding with wild type or DANA human IgG1. Hence, we could not determine if DANA mutations in F9H4 would affect the binding to FcRn because the assay did not work for wild-type Fc. On the other hand, in the literature, this information is already known for other antibodies. For example, it was reported in Shields *et al*, 2001 (PMID 11096108) that DANA mutations in human IgG1 did not reduce or increase significantly the binding affinity to human FcRn by an anti-IgE antibody (clone E27). We included that information and citation on page 6, lines 12-13. We have also used another DANA-mutant antibody in previous studies (e.g. PMID 34359073), and thereby we are comfortable with that approach.

- In figure 3C, the authors show a flowcytometry-based tumor killing assay with 7-AAD. 7-AAD was added after the 4 hour incubation of NK+A549 cells, and dead A549 cells without NK cells present were subtracted. Hence,

the overall percentage killing then includes both dying NK cells and A549 cells. How is this then a true measure of A549 killing? Why not use a more robust tumor killing assay with luciferase or radioactive chromium? Why not use CFSE to specifically stain target cells, allowing live and dead A549 cells to be quantitatively distinguished. Was a positive control taken along for 100% tumor killing?

Additionally, from text, figure and legend it is not clear that there was gating on NK cells, hence show gating strategy for the NK cell effector function assays in supplemental and elaborate on gating in figure legend.

As requested, we included the gating strategies in Figure S11A-I for the NK cell effector function assays that are shown in Figure 3A-C. In the NK cell-mediated killing assay with 7-AAD (Figure 3C), NK cells were gated out by using CD56 as a marker (Figure S11E-I) and we included that information in the Figure 3C legend. Unfortunately, we do not have the scintillation counter or radioactive materials license to perform the “gold standard” assay with Cr-51. Hence, we had to use an alternative method. We used 7-AAD because, conceptually, it does not “touch” the target cells during co-culture because it is added after, whereas CFSE stains them prior to co-culture. Luciferase would require that the target cells would be engineered to express it, but we wanted not to modify genetically the target cells to have a more translatable system. We did not use a positive control for 100% tumor killing, but we included representative histograms that show the differences (Figure S11I). Since we removed NK cells from the analyses by using CD56 and the differences in 7-AAD can be visualized in histograms, we are confident that this flow cytometry assay, although not the best method if compared to Cr-51, still provided the answer that there is an increase in ADCC by F9H4+cetuximab.

The author use the phrase 'cetuximab-tagged tumorcells'. This is not a very immunological statement. Immunologists would say cetuximab-opsonized tumorcells. In general, the manuscript would benefit from a critical review by an immunologist.

We replaced the term “tagged” for “opsonized”. The manuscript was critically reviewed by a colleague, Sergio Lira, MD, PhD, who is a Professor of Medicine, specialist in mucosal immunology, and former director of the Immunology Institute at Mount Sinai (New York). We thanked Dr. Lira in the Acknowledgments section.

- Test other tumor types in tumor killing assays (luciferase or chromium based) to examine extent of effectiveness. Especially since it only worked with half of the EGFR antibodies tested. Does it work better in solid tumors where there is more NK cells infiltration like, e.g. low-stage, high-grade?

To address this concern, we tested F9H4 and cetuximab in two colorectal cancer cell lines that differ in EGFR expression. HCT-116 expresses relatively high levels of EGFR and cetuximab promoted NK cell degranulation and F9H4 synergized with cetuximab (Figure S20A-B). Such effect was restricted to cetuximab with wild type Fc, whereas cetuximab-GAALIE triggered a degree of NK cell degranulation that was not further increased by F9H4 (Figure S20B). On the other hand, RKO expresses lower levels of EGFR if compared against the expression by HCT-116 (Figure S20A). Cetuximab or cetuximab-GAALIE, regardless of the combination with F9H4, did not trigger NK cell degranulation against RKO (Figure S20B). Collectively, these results may indicate that for F9H4 to promote the ADCC by cetuximab, the target cells need to express relatively high levels of EGFR.

We also assessed if F9H4 would synergize with another antibody in a melanoma metastasis model. We combined F9H4 with 7C6-hlgG1, which is a MICA/B antibody that promotes NK cell-driven immunity by means of NKG2D and CD16a according to Ferrari de Andrade *et. al*, 2018 (PMID 29599246). However, we observed that 7C6-hlgG1 by itself greatly reduced the number of B16F10-MICA metastases in the lungs of hFcR mice. F9H4 alone also caused similar effects, likely because MICA is a foreign antigen to the murine immune system that develops antibodies against it. These endogenous antibodies were characterized in PMID 29599246. Likely for these reasons, we did not observe a synergism between F9H4 and 7C6-hlgG1 in inhibiting metastases (Figure S21A). On the other hand, we detected increases in the NK cell absolute numbers in the blood of hFcR mice that were subjected to the metastasis model and treated with both antibodies (Figure S22B). Furthermore, surface CD16a expression levels were increased in blood NK cells of mice that were treated with F9H4, 7C6-hlgG1, or both (Figure S22C). Although these results are not definitive for the efficacy of this antibody combination, they are promising by indicating the possibility that F9H4 could work in combination with antibodies against a cellular stress-related surface protein. We want to further study this combination therapy in additional tumor models in follow-up projects and thereby we had to stop with the B16F10-MICA model because this is a new direction that would cause a major extension of the current manuscript.

- It has been reported that different anti-CD16 antibodies (e.g. CD16, 3G8, B73.1 and MEM-154) influence NK cell expansion rates (Kim et al, 2023, Scientific reports). Does F9H4 affect NK cell expansion?

We performed NK cell proliferation assays through CFSE dilution by flow cytometry. However, CD16a is expressed by CD56^{dim} NK cells, which have lower rates of proliferation compared to CD56^{bright}. Having this caveat in mind, we would like to inform that we did not detect increases in proliferation by F9H4 or 3G8 (Figure S12A-B). In these assays, fresh NK cells were cultured for seven days after isolation from volunteer donors and then labeled with CFSE and treated with the antibodies, which was followed by analyses on days 3 and 5 after treatment; we detected only one round of proliferation after five days from treatment (Figure S12A-B). We are not disputing previous studies that showed NK cell proliferation in response to 3G8, because in our experiments the antibodies were not cross-linked and the cells were also treated with cytokines. On the other hand, these results are in line with the argument that F9H4 is a non-agonist.

- In figure 4F, there is a strong significant difference between Isotype+NK1.1 and antibodies+NK1.1, suggesting that the antibodies also exhibit non-NK-cell-dependent anti-tumor activity. Please elaborate.

CD16a is also expressed by monocytes and macrophages. Although F9H4 did not increase surface CD16a expression levels in blood monocytes and intratumoral macrophages, we hypothesized that the latter is needed for optimal biological *in vivo* activity of F9H4+cetuximab. We applied macrophage depletion through intratumoral injection of clodronate liposomes, a method that was reported recently by Osorio *et al.*, 2023 (PMID: 37977147). Macrophage depletion interfered with the anti-tumor activity of F9H4+cetuximab in the LLC1-hEGFR model in hFcR mice (Figure 4I). Despite that, *in vitro* experiments with human monocyte-derived macrophages from volunteer donors did not reveal a synergism between these two antibodies; rather F9H4 slightly lowered the ADCP levels that were triggered by cetuximab (Figure S14). However, A549 is relatively resistant to phagocytosis due to expression of “do not eat me” signals such as CD47 (PMID 35229723). Our results are in line with that, because we also only detected low levels of ADCP (Figure S14). Since F9H4+cetuximab required macrophages to optimally inhibit tumor growth but did not induce ADCP to a greater extent than cetuximab alone, our results highlight the importance of the combination of *in vitro* and *in vivo* experiments as well indicating the possibility that F9H4 could induce macrophage immune functions other than only phagocytosis, such as cytokine production or antigen presentation.

- The authors introduce a GAALIE mutation to enhance the Fc tail of cetuximab. However, there are papers stating that this mutation reduces binding affinity to all FcγR receptors compared to wildtype (Patel *et al.*, J Immunol, 2023) and that ADCC is impaired with GAALIE Fc mutations. However, glycosylation patterns are important for the enhanced binding. Elaborate more on this in discussion.

As requested, we elaborated more on this in the Discussion (page 18, lines 1-5). However, we searched for this publication (Patel *et al.*, J Immunol, 2023) in Pubmed and Google but we did not find it. For clarification, the GAALIE mutations have the opposite effect, they increase the binding affinity between the hIgG1 Fc and Fc gamma activating receptors. For example, GAALIE mutations in anti-influenza antibodies increased the binding affinities to the Fc gamma activating receptors and provided superior protection in viral respiratory infection models (PMID: 33032297). Similar results were obtained for SARS-COV-2 mAbs (PMID: 34547765). Here, we demonstrated that the binding of cetuximab to CD16a-F176 can only be detected by ELISA when we used cetuximab-GAALIE (Figure S19C). Furthermore, cetuximab-GAALIE was more efficient than cetuximab with wild type hIgG1 Fc in triggering ADCC (Figures 5B-C, S21B). Therefore, GAALIE-mutant Fc has higher binding affinity to CD16a and better induces NK cell effector functions.

- In the discussion, “Therefore we reason that the CD16a cleavage occurs not only in the immune synapse and ..., page 13, lines 24-25” Please elaborate more, as data suggest that shedding happens in immune synapse after cetuximab treatment.

We elaborated more as requested (page 15, line 23 onwards).

- Discussion, “However, such synergism was not observed for zalutumumab or nimotuzumab”. Please reason why this is not the case.

To understand better the synergism between F9H4 plus other EGFR antibodies, we performed additional ELISA whereby we compared the abilities of these antibodies to bind recombinant human EGFR protein. Nimotuzumab was not as efficient as cetuximab, necitumumab, or zalatumumab in binding EGFR in the ELISA (Figure S18A). These results help explain why F9H4 did not synergize with nimotuzumab, which had a weaker binding to EGFR. On the other hand, we think that F9H4 did not synergize with zalutumumab because its epitope may be different to the one for cetuximab. However, we acknowledge that the currently literature is scarce in regards to the zalatumumab epitopes.

Please discuss how binding to CD16b could potentially act as a sink when used in human patients. The mouse models lack this receptor since there are no human PMN present.

We thank the Reviewer for this new insight. As requested, we elaborated on this in the Discussion (page 17, lines 1-8). Also please note that hFcR mice have murine neutrophils with human CD16b expression (PMID: 22474370).

Minor points

- On page 6, line 24, it is stated that F9H4 is an agonist, but in line 19 on the same page it is stated that F9H4 is a non-agonist.

We apologize for this mistake. F9H4 is not an agonist. We deleted that sentence.

- Page 11, lines 9-10 do not read well

We edited the text in these lines to improve clarity: *“Since CD16a binds the hlgG1 Fc, which is conserved, we hypothesized that F9H4 is applicable for combination with other antibodies beyond cetuximab.”*

- Figure 6G and S14, color legend is missing

We corrected that in both figures.

- The manuscript is missing an abbreviations list

We included an abbreviation list at the end of the manuscript file, just before the main figures.

Reviewer #2 (Remarks to the Author)

The manuscript by Bortoleti et al. introduces a novel pharmacological approach to inhibiting CD16 shedding, thereby enhancing anti-tumor activity. The authors describe the development of F9H4, an engineered antibody that binds CD16a/b via its Fab fragment and allosterically prevents CD16 cleavage. They demonstrate that F9H4, when combined with Cetuximab, promotes antibody-dependent cellular cytotoxicity (ADCC) both in vivo and in tissue samples from lung cancer patients. These findings are innovative, clinically relevant, and thoroughly discussed, including the limitations of the study. While the results are generally compelling, the following points should be addressed:

1. Statistical analysis concerns: For example, in Figure 1B, the authors employed ANOVA, which assumes a normal distribution. However, there is no evidence to support this assumption, and the number of observations appears insufficient. A non-parametric alternative, such as the Kruskal-Wallis test, would be more appropriate. Furthermore, for some flow cytometry results, the normal distribution of CD16 expression is clearly violated (e.g., Figure S1, influencing Figures 1C, 1D, and Figure S2, influencing Figure 1F). As such, statistical reassessment is warranted in several figures, including Figures 1E and 1H.

We thank the Reviewer for these statistical insights and we made every effort to address them by including a statistician (Dr. Diniz) in our team and who re-analyzed the data through nonparametric tests. We are glad to inform that the parametric tests in Figures 1B-I were replaced for Rank-Based Multiple Test Procedures and Simultaneous Confidence Intervals (Konietschke et. al, Electron J Statist 6: 738-759, 2012).

We also would like to clarify the concern with flow cytometry histograms. They do not always show normal distributions because not all NK cells or macrophages express CD16a. Hence, the surface CD16a levels display a pattern that flow cytometrists usually call “bimodal”. Because of bimodal distribution, we analyzed the percentage of CD16a⁺ NK cells and macrophages in Figures 1C-F and I. On the other hand, MOLM13 was genetically engineered through lentiviral transduction that was followed by flow cytometry cell sorting for isolation of the transduced cells, and thereby we analyzed the mean fluorescence intensities (MFI) (Figures 1H and S6). Finally, CD16b expression was bimodal in neutrophils and although analyzes of percentage revealed increase by F9H4, the dose-response effect was only apparent when analyzing MFI (Figure 1G, S4B). Therefore, these histograms show the patterns of expressions in the cell populations.

Additionally, in Figures 3A-D, normal distribution cannot be assumed, making three replicates insufficient. The statistical test used for Figures 3A and 3B is not mentioned. Some figures also lack statistical significance markers, making it challenging to interpret or support the authors' claims. The authors should revisit their statistical analyses across the manuscript and update their methodology where needed.

Our statistician collaborator (Dr. Diniz) revisited the statistical analyses and switched the parametric analyses in Figures 3A-D to the nonparametric test. Hence, we indicated in the Figure 3 legend that we used “Rank-Based Multiple Test Procedures and Simultaneous Confidence Intervals”. We also included the statistical significance markers. We edited the Methodology to indicate the test used accordingly.

2. Inconsistency in measurement variables: In Figures 1F and 1G, the authors performed similar experiments on two different cell types but presented different metrics: percentage of CD16-expressing cells in Figure 1F and MFI in Figure 1G. A similar discrepancy exists in Figure 1E. The authors should clarify why they chose to present different variables for comparable experiments to ensure consistency and transparency.

The differences in metrics (% vs MFI) occur primarily because of unimodal or bimodal patterns of surface CD16a/b expressions, as mentioned above. In Figure 1F we used % because the CD16a expression pattern is bimodal in macrophages (Figure S3). In Figure 1E, CD69 expression is unimodal whereas CD16a expression is bimodal, which is why the first was depicted as MFI and the latter as %. We now show histograms that represent the CD69 and CD16a expression patterns (Figure S2D). On the other hand, although the % CD16b expression was bimodal in neutrophils (Figure S4B), we used MFI in the main figure because the MFI values enabled to see the dose-response effect (Figure 1G).

3. Negative CD16 values in Figure 1B: The ELISA results in Figure 1B show CD16 levels below zero, which is unexpected for this assay. Potential causes include over-subtracted background, sample preparation errors, or extrapolation of values below the detection threshold of the standard curve. The authors should clarify their methodology and explain these negative values to ensure accurate data interpretation.

The negative values were caused by the interpolation of absorbance values below the detection limit of the standard curve. We added that info in the Figure 1B legend. It is our interpretation that such negative values mean “not detected”. We prefer not to replace the negative value data with “not detected”; rather, we show the actual values so the readers can better understand the result. Furthermore, we included in Figure S1 a cartoon that illustrates the assay.

4. CD16 internalization: While the authors assessed CD16 expression on the cell surface and in the medium, they hypothesize potential internalization of CD16. To validate this hypothesis, the inclusion of internalization assays, such as intracellular staining, would be beneficial. These experiments could measure CD16 levels following F9H4 treatment, with and without tumor cell challenge, and provide mechanistic insights into cleavage-independent regulation of CD16.

We thank the Reviewer for this interesting idea and we performed experiments to better understand the F9H4-induced CD16a internalization. We analyzed the NK cell surface CD16a expression levels in cold and warm temperatures. The reason for that setting is that F9H4 should not induce CD16a internalization in NK cells that were metabolically slowed down when kept at 4 °C degrees. Indeed, that was the case (Figure S13A). On the other hand, at 37 °C degrees, we observed a small reduction in surface CD16a expression levels by F9H4 in the absence of PMA. Such reduction was statistically significant but much less than the reduction by PMA (Figure S13B). Therefore, the levels of CD16a internalization by F9H4 are very low and for that reason, we did not follow up with intracellular staining.

5. Subpopulation analysis: CD16 expression differs between NK cell subpopulations (CD56Dim/CD56Bright). Evaluating the effects of F9H4 on these distinct subpopulations would provide a more comprehensive understanding of its impact on NK cell function.

We would like to clarify that CD56^{bright} NK cells do not express CD16a. The absence of CD16a in CD56^{bright} NK cells is not by cleavage, but by low mRNA levels (PMID 31801909). Therefore, F9H4 targets CD56^{dim} NK cells, which are the most abundant NK cell subpopulation in the human blood.

6. Fresh vs. activated NK cells: The in vitro assays utilized activated NK cells, as stated in the Methods section. However, receptor expression differs between fresh and activated NK cells. Assessing the impact of F9H4 on fresh NK cells would provide additional insights. Specifically, this analysis might help explain the unexpected results in Figures 4D and 4E, where circulating NK cells (likely fresh) seem affected by F9H4, whereas tumor-infiltrating NK cells (likely activated) show no reduction in CD16 expression—or even higher levels—in the absence of the antibody.

As requested, we performed CD16a shedding assays with fresh NK cells from volunteer donors. To do so, we did not purify NK cells but used whole PBMC. These assays confirmed that F9H4 inhibited the CD16a shedding by fresh NK cells (Figure S5A). In these experiments, we analyzed MFI because the NK cell marker, CD56, is

shed after PMA too but we included T cells, neutrophil, and monocyte markers that allowed us to also analyze CD16a/b expressions in these cells or gate them out when analyzing NK cells; F9H4 also inhibited CD16a and CD16b shedding by fresh monocytes and neutrophils, respectively (Figure S5B-C).

7. Figure 4D-H: Alignment of results: As mentioned in the previous section, Figure 4E indicates that CD16 expression on tumor-infiltrating NK cells and macrophages is not affected by F9H4 or Cetuximab. This observation appears to contradict earlier results, where CD16 was significantly modulated by the antibody. The authors should discuss and explain this inconsistency, providing insights into the biological or experimental reasons that might underlie this discrepancy.

For F9H4 to inhibit the CD16a/b shedding, the shedding needs to be induced. We used PMA, co-culture with tumor cells, and cetuximab to induce the CD16a shedding by NK cells, and the first for macrophages *in vitro*. On the other hand, the CD16a shedding *in vivo* needs to be triggered only by physiologically relevant stimuli. Cetuximab induced the CD16a shedding by blood NK cells in tumor-bearing hFcR mice and F9H4 inhibited this shedding (Figure 4D). However, blood monocytes did not shed CD16a in these same mice and thereby we could not observe an impact by F9H4 in the CD16a expression in blood monocytes (Figure 4G). On the other hand, F9H4 did not prevent the CD16a downregulation by tumor-infiltrating NK cells and macrophages (Figure 4E, H). These *in vivo* data indicate that in the tumor microenvironment CD16a expression is likely modulated by not only cleavage but other processes, such as internalization. Receptor engagement commonly causes internalization, to facilitate intracellular signaling. Furthermore, cytokines in the tumor microenvironment may modulate CD16a expression at the mRNA levels. We included these information in a new paragraph in Discussion (page 17, lines 10-19).

8. NK cell validation: The authors should clarify how they validate NK cell integrity after isolation and during the experiments. While the Methods section describes NK cells as CD45+/CD56+ for identification, some figures (e.g., Figure 3) suggest NK cells are considered CD45+ alone. Providing the gating strategy used in the *in vitro* experiments would enhance clarity and reproducibility.

As requested, we included the flow cytometry-based validation of NK cell purity (Figure S2A) as well as the gating strategy for the *in vitro* assays in Figure 3A-C (Figure S11A-I).

Reviewer #3 (Remarks to the Author)

To the Authors

In their manuscript entitled “Discovery of a monoclonal antibody that binds CD16a to inhibit the shedding and promote anti-tumor immunity” Bortoleti et al describe the generation and functional characterization of a novel human FcγRIII (CD16 antibody), named F9H4. This FcγRIII directed antibody is shown to inhibit shedding of the receptor and to enhance ADCC by human NK cells. *In vivo*, co-administration of F9H4 enhanced the efficacy of two (cetuximab and necitumumab) but not of two other (zalutumumab and nimotuzumab) EGFR antibodies in a mouse model of human EGFR transfected LLC1 syngeneic tumor cells in hFcR mice.

Overall, this is an interesting manuscript that describes an approach to inhibit the well-described shedding of human FcγRIII to enhance the efficacy of EGFR antibodies. However, many important issues are not properly addressed

Major:

1. The title claims that the CD16 antibody “promotes anti-tumor immunity”, which would suggest a tumor-directed adaptive immune response. This, however, is not shown and cannot be concluded from the presented data. The term “immunity” is often used too loosely (e.g. page 2, line 23; page 7, line 25; page 13, line 13).

Dear reviewer, thank you for the efforts to improve the scientific quality of our manuscript. We edited the title and main text to be more specific throughout the manuscript. For example: “*NK cell-mediated cytotoxicity and interferon-γ production*” rather than “*NK cell-driven immunity*” in the first sentence of the Discussion as requested.

2. Throughout the manuscript, the authors emphasize the role of NK cells and ADCC for the therapeutic efficacy of EGFR antibodies in patients. However, potential contributions from other FcγRIII expressing cells are widely neglected. For example, macrophages mediate ADCP and PMN ADCC via EGFR antibodies. The impact of F9H4 on these effector mechanisms could have easily been addressed, at least *in vitro*.

As also explained above, we performed ADCP assays through CFSE uptake, as we did in a previous study (PMID 34359073). However, we did not detect increases by F9H4 on ADCP against A549 *in vitro* by human monocyte-derived macrophages (Figure S14). In addition to that, we applied macrophage *in vivo* depletion in the LLC1-EGFR model in hFcR mice and discovered that clodronate liposomes, which were administered by intratumoral injections, interfered with the inhibition of tumor growth by F9H4+cetuximab (Figure 4I). Therefore, macrophages appear to play an important role for the *in vivo* activity of F9H4+cetuximab. For clarification, we did not perform neutrophil effector function assays because to the best of our knowledge the roles of CD16b for anti-tumor immunity are not well delineated yet, but that definitely would be an interesting new direction.

3. Only EGFR antibodies were used to demonstrate the impact of F9H4. Seeing that the approach would also work with other tumor targeting antibodies would greatly increase the impact of the current manuscript.

We agree with the Reviewer and this has been one of the most challenging aspects of our project. Here, we performed *in vivo* experiments whereby hFcR mice were inoculated intravenously with B16F10-MICA cells, which form metastases in the lungs. We treated the mice with mAb combination to assess if F9H4 would synergize with 7C6-hlgG1, which is a mAb that inhibits the shedding of MICA and MICB (MICA/B) that are NKG2D ligands (PMID: 29599246). We chose this approach because MICA/B are expressed by different cancer types in response to cellular stress pathways and 7C6 is broadly applicable. We observed that 7C6-hlgG1, by itself, greatly inhibited the development of B16F10-MICA metastases in the lungs. F9H4 alone also inhibited metastases likely because hFcR mice develop endogenous antibodies against B16F10-MICA and thereby F9H4 should promote the Fc effector functions of endogenous antibodies. Although these experiments did not demonstrate lower metastasis burden in mice that received both antibodies simultaneously compared to F9H4 or 7C6-hlgG1 alone, we detected increases in the NK cell absolute numbers in the blood in the combination treatment group (Figure S21A-B). Furthermore, blood NK cells had higher expression of CD16a in hFcR mice that were treated with F9H4, 7C6-hlgG1, or both (Figure S21C). We did not detect differences in NKG2D expression levels in blood NK cells (Figure S21D). In summary, we consider these new findings to be encouraging because they indicate the possibility that F9H4 may synergize with a new antibody broadly applicable to cancer immunotherapy in the future. We want to follow up in a new study whereby we can apply F9H4+7C6 to different cancer models such as leukemia, but we had to stop here in the B16F10 model because such a new direction would require a more significant extension of the current manuscript.

4. No explanation is provided why some EGFR antibodies benefit from co-administration of F9H4, while others do not. For example, cetuximab and zalutumumab are very similar EGFR antibodies.

As also mentioned above, we performed additional ELISA to compare the abilities of these antibodies to bind recombinant human EGFR protein. Nimotuzumab was the weakest binder, whereas cetuximab, necitumumab, and zalatumumab bound recombinant EGFR protein with similar efficiencies (Figure S18A). These new data help explain why F9H4 did not synergize with nimotuzumab. However, they do not explain why F9H4 did not synergize with zalutumumab. We think that the lack of synergism is driven by zalutumumab epitope, as mentioned in the Discussion. The epitopes recognized by cetuximab and zalatumumab are similar but may not be identical. Unfortunately, the literature is currently scarce in regards to how zalutumumab epitopes differ from cetuximab epitopes, thus limiting our abilities to explain why F9H4 did not work in synergism with the first.

5. In Figure 1, PMA is used to trigger shedding of FcgRIII, but the presentation of data is difficult to understand. Apparently, effector cells (NK cells C + D, macrophages in F and PMN in G) were stimulated with PMA and CD16 expression was measured. Why are results presented as "% expression" in C and F, but as "MFI" in G? Apparently, CD16 shedding was prevented in NK cells, but in macrophages and PMN CD16 expression was raised by F9H4? How is staining with F9H4 affected by shedding of the receptor? Here, representative immunofluorescence blots would be informative. Apparently, the Legends in C and D are mixed up. In I, CD16 expression goes up upon incubation with F9H4-DANA? How were percentages calculated ?

Dear reviewer, we apologize for this confusion. F9H4 does not increase CD16a expression, but it retains CD16a on the surface. In Figures 1D, 1F, 1G, and 1I the graphs give the impression that F9H4 is increasing the CD16a expression levels, but these leukocytes were all treated with PMA to induce the shedding. Hence, in these assays, CD16a/b shedding are being triggered by PMA and that enabled us to titrate the antibody doses. On the other hand, in Figure 1C we titrate the PMA doses and thereby it is possible to visualize that F9H4 prevents the downregulation that otherwise would be induced by the shedding. We confess that find it a bit difficult to improve clarity in Figure 1 because on one hand we titrate the PMA dose or analyze in different times to show how PMA induces CD16a shedding (Figures 1C, E, and H). On the other hand, we also titrated the mAb dose in PMA-

treated leukocytes to show how F9H4 inhibits the shedding in a dose dependent manner (Figures 1D, F, G, and I). We added titles to the graphs to hopefully enhance clarity.

6. Several of the sub-figures (e.g. Figure 1A and E, Figure 2D and H) do not show SEM although apparently more than one experiment was performed. The term “triplicates” should be specified whether technical or biological triplicates are shown.

Figures 1A, 2D, and 2H do not show error bars because the ELISA was performed with one replicate per data point. Figure 1E does show error bars, but perhaps the Reviewer was referring to Figure 1G. This figure consist of mean \pm standard error of technical triplicates, but the error bars are too small to be seen. We stated in Methodology that in these assays triplicates refer to the amount of wells that were plated/used. In general, we performed the *in vitro* experiments at least three times and each one of them in at least triplicates.

Minor:

1. The clarity of the Figures (in particular Figure 1) could be improved by adding small cartoons describing the set-up of some of the more complex assays. Also, a graphical abstract of the manuscript would be helpful.

Figure 1 was already a relatively large figure, so we included in the Supplementary File the illustration for the complex assay with soluble CD16a detection (Figure 1B, S1). We also included a Graphical Abstract.

2. Throughout the manuscript, the antibody 3G8 is often described as an anti-CD16a antibody (e.g. line 15-16), which is not correct. 3G8 is pan-CD16, recognizing both CD16a and CD16b. Also the title is misleading since F9H4 is pan-CD16 and not CD16a specific.

Thank you for highlighting this mistake. We corrected that throughout the manuscript and edited the title.

3. Apparently F9H4 binds to the D1 domain of FcγRIIIa (Figure 2) and inhibits cleavage, which is supposed to occur between D1 and D2. It would be interesting to model the F9H4 binding site to FcγRIII – similar to Figure 1A. Do other commonly available CD16 antibodies also inhibit shedding?

We would like to clarify that the cleavage occurs in the stalk that is in-between D2 and the transmembrane domain (PMID 25816339). We included a Graphical Abstract that illustrates the binding of F9H4 to CD16a. A mAb that binds CD16a and inhibits the shedding was not described in any prior study. Furthermore, our data with the D1 mAbs that were raised by immunizing mice with recombinant D1 protein indicated that not any D1-targeting mAb would inhibit the CD16a shedding (Figure 2I). F9H4 is not commercially available, but we would be happy to share it with the scientific community.

4. On page 11, line 9 other EGFR antibodies (nimotuzumab, necitumumab zalutumumab) are described as “biosimilars” of cetuximab, which is not correct.

We deleted the word biosimilars and edited the text as requested: “*We analyzed other three antibodies that target EGFR, are hlgG1, and have been tested in patients: 1) Zalutumumab...*”

5. Numerous parts of the manuscript require editorial work. E.g.

a. Page 2 line 11: “specificity to hundred types of proteins”.

We edited the text as requested: “*... is involved in the cleavages of many different types of proteins.*”

b. Page 2 lines 18 – 21.

We edited the texts in these lines: “*An advantage of F9H4 is that it targets the substrate (CD16a) and spares the protease (ADAM17), and that unique feature enabled in vivo experiments in Fc gamma receptor-humanized (hFcR) mice. We discovered that F9H4+cetuximab inhibited the growth of murine lung carcinoma in hFcR mice and such effect was mediated by NK cells and macrophages.*”

c. Suppl. Page 6, line 18: “staining with 3G8 for CD16a expression”. PMN express CD16b.

We apologize for this mistake, and we correct that.

6. Often the Legend of Suppl. Figures state that “data represent three independent experiments”. Here, “means \pm SEM” should be depicted to give the reader an impression about the variation of results.

In agreement with the Reviewer, we noted that many of the ELISAs in Supplementary Figures had no information about replicates. Our laboratory performs ELISAs with only one well per antibody concentration, and thereby we

added to the Figure Legends that each data point is one observation and the whole graphs represent three independent experiments.

REVIEWER COMMENTS

Reviewer #1 (Remarks to the Author):

The manuscript has significantly improved in the revision.

Reviewer #2 (Remarks to the Author):

No further comments. The authors responded to all criticisms

Reviewer #3 (Remarks to the Author):

The authors provide an extensively revised manuscript, in which important biological questions are still not properly addressed:

1. Why is NK cell mediated ADCC by cetuximab enhanced, while that by zalutumumab is not affected? Both EGFR antibodies bind to overlapping epitopes of EGFR and mediate similar levels of ADCC. If inhibition of FcγRIII shedding is the relevant mechanism of enhanced ADCC by F9H4 why is it so highly dependent on the EGFR fine epitope? How about additional EGFR antibodies?
2. Why did the authors not use antibodies against other target antigens like HER-2/neu or CD20 to demonstrate that F9H4 does not only work with cetuximab?
3. With macrophages as effector cells, apparently only A549 cells were tested as target cells *in vitro*. In the published literature other tumor cell lines with higher EGFR expression levels were shown to be phagocytosed by macrophages in the presence of EGFR antibodies. Again, antibodies against other target antigens (please see also above) could be used to demonstrate broader applicability of this novel approach.

Authors' Responses to the Reviewer #3

Dear Reviewer, thank you for re-reviewing our manuscript and, again, acknowledging the innovation in our approach. We are sorry that our previous responses were not fully satisfactory, but we have now performed additional experiments to specifically address these three items. Our responses are as follows:

1. Zalutumumab and additional EGFR antibodies. We understood that this critique was particularly relevant to explain Figure 5A, which shows that F9H4 synergized with cetuximab and necitumumab but not with nimotuzumab or zalutumumab; hence, we tested the combination of F9H4 with four antibodies against EGFR. Since the first resubmission, we have shown this ELISA in which nimotuzumab was the weakest binder (Figure S18A). To complement, now in this second resubmission we included additional analyses with the LLC1-hEGFR cell line. We compared the abilities of cetuximab and zalutumumab to bind LLC1-hEGFR; the binding was revealed with anti-human IgG secondary antibody. We found that zalutumumab was a weaker binder to LLC1-hEGFR cells at intermediate antibody doses (Figure S18B). Therefore, our explanation for why F9H4 synergized with cetuximab but not with zalutumumab *in vivo* is that cetuximab better opsonized LLC1-hEGFR cells. Finally, it is also worth highlighting that F9H4 synergized with the two FDA-approved antibodies (cetuximab and necitumumab) (Figure 5A).

2. Antibodies against other target antigens. To address that concern, we performed ADCC and ADCP assays. We detected small but significant increases by F9H4 in NK cell degranulation and macrophage-mediated phagocytosis of rituximab-opsonized Raji cells (Figure S14B and D). On the other hand, we did not detect increases by F9H4 in ADCP against rituximab-opsonized OCI-Ly10 or SU-DHL4 cells (Figure S14D). We also performed degranulation assays against trastuzumab-opsonized SKOV3 cells but did not detect a synergism between F9H4 and anti-HER2 antibody in promoting NK cell degranulation (Figure S14C). Collectively, these new data show that the ability of F9H4 to promote ADCC/P is modulated by multiple factors such as target cells, target antigens, and target cell-opsonizing antibody, but we were able to document a synergism between F9H4 and rituximab in ADCC/P against Raji cells.

Furthermore, we performed *in vivo* experiments in which we combined F9H4 with 7C6-hIgG1 in the LLC1-MICA model. Since the first resubmission, we have shown how F9H4, 7C6-hIgG1, and both in combination inhibited B16F10-MICA metastases in Fc gamma receptor-humanized mice; however, these antibodies were similarly effective, which did not enable us to demonstrate the synergism (Figure S21A-D). We attempted again,

but this time in the LLC1-MICA subcutaneous tumor model; however, we found similar results. F9H4, 7C6-hlgG1, and F9H4 plus 7C6-hlgG1 inhibited tumor growth. Although the effects in the LLC1-MICA model were weaker if compared against the effects in the B16F10-MICA model, the LLC1-MICA tumor growth was significantly inhibited but to a same extent among the three treatment groups (Figure S21E-F).

We agree with the Reviewer that it would enhance the study if we could demonstrate that F9H4 also works synergistically with antibodies against other target antigens, and indeed we have tried with rituximab and trastuzumab *in vitro* and 7C6-hlgG1 *in vivo*. But so far cetuximab provided the greatest synergistic effects. Despite this relatively narrow scope in the partnering antibody, our article still is expected to cause a profound impact on this research field because F9H4 is the first-in-class antibody inhibitory of CD16a/b shedding. We comprehensively characterized this antibody and provided evidences that CD16a/b shedding is physiologically and clinically relevant throughout the manuscript. These data provided solid foundation that CD16a/b shedding is a “therapeutic target” that can be “hit” by F9H4, to enable future works that may leverage F9H4 at its full potential for cancer immunotherapy. Our study is the first to describe F9H4 and we would be delighted to share it with the scientific community to identify and optimize further combinations with antibodies against other target antigens.

3. ADCP assays. We apologize for not attempting to demonstrate ADCP with more target cells in the previous resubmission. As explained above in “2”, we have conducted more ADCP assays, but this time with rituximab-opsonized lymphoma cells. For clarification, the reason for using lymphoma cells was that A549 cells are adherent, whereas those lymphoma cells grow in suspension. We assumed that it would be easier to detect phagocytosis with non-adherent target cells. Our assumptions were likely correct, because as it can be seen in Figures S14D rituximab induced robust phagocytosis in these experiments, thus validating the assays. On the other hand, we did not attempt to perform ADCP assays with anti-HER2 antibodies because SKOV3 is adherent. We are not saying that it would not be possible to conduct ADCP assays with adherent target cells, but in our experience it was easier to detect ADCP with non-adherent target cells, which is why we focused on lymphoma cells.

Collectively, we hope that these new data and clarification will satisfactorily respond to the Reviewer’s concerns. We thank again for this careful evaluation of our manuscript and we appreciated the opportunity of improving our work!

REVIEWERS' COMMENTS

Reviewer #3 (Remarks to the Author):

Dear Authors,

thank you very much for your efforts to address my concerns and to broaden the applicability of your findings by testing additional antibodies and target cell lines.

Overall, I am not impressed by the new results and still wonder what is special about EGFR and cetuximab. To my understanding inhibition of FcγRIIIa shedding should have stronger effects also with other antibodies and additional cell lines.

Response

Our study provides a comprehensive characterization of the monoclonal antibody (mAb) F9H4, which inhibits CD16a and CD16b shedding, in *in vitro* assays and tumor models that revealed new mechanistic insights into Fc receptor biology. Although we primarily focused on the combination between F9H4 and cetuximab in cancer immunotherapy experiments, other relevant combinations can also be explored in future studies. There are multiple antibodies that opsonize target cells in cancer and viral infections, and thereby F9H4 has the potential of being broadly applicable. Therefore, our study used F9H4+cetuximab to provide proof of concept. Furthermore, F9H4 is a key intervention that enabled us to demonstrate that CD16a and CD16b shedding is physiologically relevant in the mouse models and clinically relevant in the patient-derived tumor samples; we also revealed, in parts, the mechanism of action of this novel mAb. Taken together, we expect that this study will cause a major impact on the basic biology and applied research fields.